# Circularly Polarized Textile Sensors for Microwave-Based Smart Bra Monitoring System

**DOI:** 10.3390/mi14030586

**Published:** 2023-02-28

**Authors:** Dalia N. Elsheakh, Yasmine K. Elgendy, Mennatullah E. Elsayed, Angie R. Eldamak

**Affiliations:** 1Department of Electrical Engineering, Faculty of Engineering and Technology, Badr University in Cairo, Badr City 11829, Egypt; 2Microstrip Department, Electronics Research Institute, Nozha, Cairo 11843, Egypt; 3Electronics and Communications Engineering Department, Faculty of Engineering, Ain Shams University, Cairo 11517, Egypt

**Keywords:** axial ratio (AR), circularly polarized (CP), breast cancer detection (BCD), microwave imaging, textile antennas, breast phantoms, tumor detection, smart bra, wireless body area network (WBAN)

## Abstract

This paper presents a conformal and biodegradable circularly polarized microwave sensor (CPMS) that can be utilized in several medical applications. The proposed textile sensor can be implemented in a Smart Bra system for breast cancer detection (BCD) and a wireless body area network (WBAN). The proposed sensor is composed of a wideband circularly polarized (CP) textile-based monopole antenna with an overall size of 33.5 × 33.5 mm^2^ (0.2 λ_o_ × 0.2 λ_o_) and CPW feed line. The radiating element and ground are fabricated using silver conductive fabric and stitched to a cotton substrate of thickness 2 mm. In the proposed design, a slot is etched in the radiating element to extend bandwidth from 1.8 to 8 GHz at |S_11_| ≤ −10 dB. It realizes a circularly polarized output with AR ≤ 3 dB operation band from 1.8 to 4 GHz and an average gain of 6 dBi. The proposed CPMS’s performance is studied both off-body (air) and on-body in proximity to breast models with and without tumors using near-field microwave imaging. Moreover, the axial ratio is recorded as a feature for a circularly polarized antenna and adds another degree of freedom for cancer detection and data analysis. It assists in detecting tumors in the breast by analyzing the magnitude of the electric field components in vertical and horizontal directions. Finally, the radiation properties are recorded, as well as the specific absorption rate (SAR), to ensure safe operation. The proposed CPMS covers a bandwidth of 1.8–8 GHz with SAR values following the 1 g and 10 g standards. The proposed work demonstrates the feasibility of using textile antennas in wearables, microwave sensing systems, and wireless body area networks (WBANs).

## 1. Introduction

A wireless body area network (WBAN) is a group of low-power, miniature, lightweight wireless sensors that monitor internal and external body health processes. WBANs help in monitoring physiological behavior patterns in humans and studying disease prevention and control [1]. The WBAN’s ultra-lightweight, wearable sensors can be either off-body or on-body [2] and could be operated in MIMO systems [3,4,5]. Numerous applications of WBANs have embraced several devices on a large scale, including brain recording, glucose monitoring, intracranial pressure monitoring, as well as breast cancer detection [2,6]. WBANs use wearable, lightweight antennas as a feature to transmit and receive signals to the human body. In this paper, a wearable bra is proposed to monitor and scan for breast cancer. This Smart Bra is presented as a component of a WBAN system such that women will not need to regularly visit a hospital [7].

Breast cancer is an asymptomatic disease that should be detected in its early stages [8]. Several common techniques in the literature for breast cancer imaging are reported in [9,10]. Although these techniques are the most common on the market, they still have some limitations [8]. Microwave detection techniques provide a safer approach as they use non-ionizing radiation. Microwave imaging relies on detecting differences in electrical properties between tumors and normal tissues within the breast [11]. The contrast ratio of dielectric properties of normal breast tissue to malignant tissue is close to 10, as reported in [12]. Hence, the microwave signal can distinguish between healthy and tumorous tissues. Several microwave-based detection systems have been reported in [9,10,13] for cancer detection, most of them focused on planar structures. The development of wearable microwave imaging systems is continuously seeking compactness, robustness, mobility, cost efficiency, and non-intrusive features. Realizing these features allows for regular, long-term assessment of cancer screening conditions, as well as the encompassing environment. Thus, it is essential to consider using comfortable and soft materials, such as textiles, for building such systems. Nowadays, textiles are becoming the first choice for fabricating and developing wearable antennas and sensors as they provide comfort to the wearer and flexibility [14,15].

Many textile fabrication technologies were discussed in [6,15,16,17,18,19,20], such as using copper tape, conductive fabric, embroidered conductive threads, or using inkjet screen printing on a non-conductive textile acting as a substrate. In [17], a study was conducted using different conductive materials and cutting assembly techniques and comparing them together. In [15,16,18], conductive fabric attached to the substrate by gluing or stitching realizes higher efficiency compared with other textile materials [17]. In [15,16,19], the embroidery technique offers very precise structures but with poor efficiency and high manufacturing costs. In [16,20], the inkjet technique is also reported. This technique is not very practical as it is challenging to realize robustness to stretching and resilience to high temperatures. Several textile antennas have been reported in the literature for medical applications [8,10,14,21,22,23,24,25,26,27,28,29,30], with few of them targeting microwave imaging applications. Most of them operate in the band from 1 to 10 GHz. Beyond 10 GHz, scattering effects occur on the skin surface [14]. However, these reports do not provide too many experimental results or thoroughly discuss the body effects on the performance of the antennas or sensors [24].

Wearable antennas can bend and stretch while in use, and accordingly, the radiation properties are varied. Thus, the CP antenna is favored in developing flexible wearable sensors and systems. CPMS microwave sensor waves contain both vertical and horizontal electric field components. Thus, receiver antennas can pick up radiation waves from a variety of angles when these waves are reflected or refracted. Moreover, circular polarization (CP) antennas reduce polarization mismatch and flexibility in terms of orientation and increased mobility. Moreover, it is a good choice for encountering multipath fading and establishing reliable channels when compared with linearly polarized (LP) antennas, especially in crowded interior environments [1,2,3,4,5]. However, only a few works focus on the CP MS wearable antennas [1,2,3,5,6,7].

Several circularly polarized (CP) antennas have been studied, including microstrip patch antennas [31], filtering antennas, and anisotropic artificial ground plane loaded antennas [32]. The truncated corner of the microstrip patch antenna in [33] is intended to produce CP unidirectional radiation. However, the coaxial probes that feed these antennas make them difficult to use in wearable applications. A wearable biotelemetric system with a tiny CP antenna integrated with band-pass filter circuits is proposed in [31]. However, it uses a rigid substrate that cannot conform to the human body. In [31], a flexible CP antenna operating at an industrial, scientific, and medical (ISM) band of 2.4 GHz is investigated. However, these wearable CP antennas have a limited axial ratio (AR) bandwidth, which could cause performance degradation due to a potential frequency offset.

In this paper, we introduce an ultra-wideband (UWB) antenna that can be utilized as a sensor in two major applications: (1) wireless body area networks (WBANs) and (2) breast cancer imaging. This paper presents a fully textile CPW monopole antenna with an operating bandwidth from 1.8 GHz to 8 GHz and circularly polarized output in the band from 1.8 to 4 GHz. The reflection coefficient, radiation pattern, gain, and SAR are simulated and recorded for the proposed antenna. The proposed work is validated through fabrication and measurements. Moreover, simulations for the proposed textile antenna with a breast phantom model are also conducted. A tumor model is also included in the breast phantom as part of the simulations presented in this paper. Simulations for the proposed antenna as a single element, as well as two-element configurations, are presented and compared with and without tumor models. The given results validate that the proposed design can fit as a microwave-based flexible sensor for breast cancer detection systems.

The proposed circularly polarized sensor is a continuation of the work in [30] to develop a Smart Bra, as shown in Figure 1. In [30], a textile-based sensor is presented as a linearly polarized microstrip antenna and has an impedance bandwidth from 1.8 to 2.4 GHz and from 4 up to 10 GHz at |S_11_| ≤ −10 dB. By enabling circularly polarized output, a new monitoring indicator, axial ratio, is added. The axial ratio is recorded in addition to the S-parameters (|S_11_| for reflection and |S_21_| for transmission). The proposed CPMS has a broad axial ratio (AR) bandwidth that allows for efficient operation in challenging conditions and in proximity to lossy, dispersive, and heterogeneous human tissues. Moreover, it is also used in analyzing data measured from the breast. This will increase the samples of collected data and will contribute to increasing detection accuracy. The proposed system is meant to help women in the early detection and continuous monitoring of breasts in the comfort of their homes.

## 2. Design and Characterization of Textile-Based Sensor

In this section, the design of the proposed circularly polarized microwave sensor (CPMS) will be presented. This includes presenting materials and design steps for the proposed sensors. In Section 2.1, the characterization of different conductive and substrates materials will be illustrated. This involves presenting the electrical properties of different materials used in sensor fabrication. In Section 2.2, the design steps for the proposed sensor will be shown, as well as its main parameters. In Section 2.3, different models and phantoms for breast tissues will be discussed. The electrical properties of fabricated breast phantoms will be measured and presented in Section 2.3. The fabricated phantoms will be used to assess the operation of the proposed sensors in proximity to breast tissues.

### 2.1. Characterization of Textile Substrates

The electrical properties of different textile fabrics, such as polyester, cotton, jeans, and crepe, are characterized using DAK-3.5-TL2 (dielectric probe station) in the range from 200 MHz–20 GHz as part of the proposed study. Different textile materials were studied as alternative materials for the substrate. The measured electrical properties are shown in Figure 2. Figure 2a shows that the real value of the relative dielectric is an almost constant behavior with frequency, while the imaginary part of the relative dielectric decreased as frequency increased, as shown in Figure 2b. The conductivity and tangent loss are shown in Figure 2c,d, respectively. In addition to the measurements in Figure 2c, cotton substrates are favored for wearable sensors with easy fabrication and integration techniques into clothes with a comparably low price point. It also offers absorption to human sweat with no allergies compared with other available substrates [8].

### 2.2. Textile-Based Circularly Polarized Antenna Design Approach

The proposed textile-based sensor in the form of a circularly polarized antenna operates in the band from 1.8 GHz to 8 GHz. The 33.5 × 33.5 mm^2^ sensor is composed of a monopole antenna fed by a coplanar waveguide (CPW) and placed on the cotton substrate. Printed CPW-fed antenna structures have several desirable characteristics, including simplicity, low profile, small size, good radiation characteristics, and low radiation loss. Additionally, using antennas with CPW-fed structures helps wireless communication devices get smaller because their uniplanar structure makes it easier to integrate with RF/microwave circuits and edge-fed connector boards [1].

The proposed antenna has a trapezoidal radiator patch with dimensions of *W*_1_ and *W*_2_, respectively, and a height of *L*_1_, as shown in Table 1 and Figure 3. The proposed design is developed into the final structure through three steps shown in Figure 3. First, the design starts with a trapezoidal monopole radiator with width *W*_1_ of 14.5 mm and *W*_2_ of 3 mm, respectively, as shown in Figure 3a. The antenna has a 50 Ω transmission feeding CPW line of width (*W_f_*) 4 mm and 0.5 mm air gaps, as shown in Figure 3. Finally, the proposed design is developed into the final structure shown in Figure 3.

The reflection coefficient response against frequency is shown in Figure 4a. The antenna acquires a bandwidth from 2.3 to 5.5 GHz at |S_11_| ≤ −10 dB. The length of the large base of the trapezoid is optimized to achieve wider bandwidth, as shown in Figure 4b. Second, two notches are introduced at the transmission feeding line, as shown in Figure 2b, to improve the impedance matching. This modification extends the operating bandwidth to 6 GHz, as shown in Figure 4a. Third, a trapezoidal-shaped slot is carved with dimensions of 10 mm and 2.5 mm, respectively, as shown in Figure 2c. This further extends the bandwidth to start at 1.8 GHz up to 6 GHz, as shown in Figure 4a. Finally, the symmetry of the trapezoidal radiator is modified by adding a trapezoidal corner, as shown in Figure 3. This helps to induce circularly polarized operation and is validated by plotting the axial ratio (AR), as shown in Figure 4c.

### 2.3. Characteristics of Breast Phantoms

The construction of human breast phantoms (glandular tissue, adipose tissues (fat), fibrous tissues covered with a skin layer) is a mandate to provide proof of concept [34]. This section will provide a review of human breast phantoms. From the literature, the most suitable phantoms for mimicking breasts are categorized into three major classes numeric phantoms, chemical phantoms (solid, liquid lab-built phantoms), and 3D printed phantoms [34,35,36,37,38,39]. Each class has its advantages and disadvantages in lifetime, frequency range, and sensitivity. In [35,36], a repository of different breast classes was obtained from MRI scans and digitized so that they can be used for 3D printed phantoms and on computer simulation technology (CST). In [37], a rubber and carbon mix is reported to be very efficient for long-term measurements. For a simple phantom construction, triton X-100 liquids have a long lifetime and can also be used by calibrating values of dielectric properties [38]. Polyethylene and agar mixtures have great mimicking values for breast tissue with broad operating bandwidth. Moreover, the ingredients and recipes are available and achievable [34].

#### 2.3.1. Numerical Models

In this section, two numerical models for breasts will be part of the simulations. The dimensions and properties of the first four layers of the phantom model used in the simulation are adopted from work presented in [8,14,21]. The given model is composed of three layers, including skin, fat, and breast grandular tissues for the breast model and one layer to represent tumor tissues [8,14,22]. A second model composed of one layer is also used with an effective dielectric constant of ε_r_ = 11 F/m. This one-layer model serves as an equivalent to the three layers breast model to expedite simulation time.

#### 2.3.2. Fabricated Models

The fabrication of breast phantoms involves creating gelatinous mixtures with specific concentrations to mimic different breast tissues, molds to build the phantoms, as well as the characterization of electrical properties of fabricated phantoms [34]. Several mixtures were presented in [36]. For phantom and tumor fabrication, a gelatinous phantom was chosen [39]. A gelatinous phantom representing the breast was composed of 150 mL corn oil, 50 mL deionized water, 30 mL neutral detergent, and 4.5 g agarose as per the recipe in [39]. On the other hand, a tumor phantom is composed of 100 mL deionized tri-distilled water, 60 mL ethanol, 1 g NaCl, and 1.5 g agarose fabrication took place in Ain Shams University’s chemistry lab in which all mixtures (breast and tumor phantoms) were held on a magnetic heating stirrer HJ-3A. The temperature of the mixtures was continuously monitored using a thermometer. After reaching the designated temperature of 80 °C, all mixtures were left to cool until a gelatinous consistency was reached. The electric properties of the fabricated models (breast and tumor) were characterized using DAK-3.5-TL2 (SPEAG’s dielectric assessment kit) in the Electronics Research Institute (ERI) in the band from 1–10 GHz, as shown in Figure 5. The measured electrical properties are shown in Figure 6, which highly matches the data reported in [34].

## 3. Simulation Results

In this section, simulation results for the proposed sensor in the air and with the breast and tumor phantoms will be presented. All simulation results are recorded using CST studio suite, high-frequency structure simulator (HFSS). In Section 3.1, the effect of changing conductor material will be studied. The conductor is an essential material for fabricating the radiator and ground plane for the proposed antennas. The simulated reflection coefficient, impedance, and gain using two conductor types will be compared in Section 3.1. In Section 3.2, the simulation of the proposed sensors with breast phantoms with and without tumors will be presented.

### 3.1. Effect of Changing Conductor Material

Simulations are conducted using two types of conductors: 0.5 mm copper clad conductor and sliver conducting fabric (sheet resistance 0.5 Ω/□). These simulations validate the similar performance of the proposed monopole antenna using a conductor fabric compared with a traditional copper clad. Both types of conductors reveal similar operations in terms of reflection coefficient, impedance matching (real and imaginary), and gain, as shown in Figure 7a–c, respectively.

The gain using both conductors is shown in Figure 7c. Gain values are almost the same in the range from 3.5 to 6 GHz with a maximum realized gain of 8 dBi by using the copper conductor, while the gain is reduced to 5 dBi by using the textile conductor fabric. The surface current is also simulated for the proposed CPW-monopole antenna, as shown in Figure 8, at 2.5 GHz, 3.5 GHz, and 5 GHz, respectively. Moreover, the current vector is also plotted in Figure 9. This helps to monitor the orthogonal current components at specific operating bands and pursue the circularly polarized operation, as shown in Figure 9.

### 3.2. Placment on Breast Phantoms

Simulations are conducted with tumor models, as in [8], with a 10 mm diameter spherical shape and placed at a distance of 60 mm in the center of the phantom. Two simulation scenarios are conducted, as shown in Figure 10a,b. The antenna is placed at a separation distance of 2 mm and filled with cotton material above the breast model with a buffer. For all simulations, |S_11_| (reflection coefficient) and |S_21_| (transmission coefficient) are recorded with and without the presence of tumors, as shown in Figure 11. The tumor size varies from 0 to 20 mm in the presence of one sensor/antenna. By increasing the size of the tumor, the resonant frequency of the monopole antenna is reduced, as shown in Figure 11. The first resonant frequency is reduced from 1.6 GHz to 1.4 GHz (0.2 GHz), and the second resonant frequency from 4.5 GHz to 4.2 GHz (0.3 GHz). Moreover, the magnitude of |S_11_| is shifted by 1 dB and 5 dB, respectively, for both bands, as also shown in Figure 11. The phase of reflection coefficient and axial ratio are also recorded and tracked for changes due to the presence of tumors in the simulated breast models, as shown later in the next paragraph. The phase of reflection of the coefficient phase is shifted by 50° due to the varying size of the tumor, as presented in Figure 11b.

In the second scenario, shown in Figure 10b (transmission measurement using two sensors placed on the breast sides), results are recorded in Figure 12. By increasing the diameter of the tumor model from 0 to 20 mm, changes in magnitude and phase of reflection and transmission coefficient are recorded in Figure 12 and Figure 13. The tumor induced a down-shift in resonant frequency by 200 MHz and 500 MHz and a change in|S_11_| magnitude by 5 dB and 15 dB, respectively, as shown in Figure 12a. The phase of reflection |S_11_| is changed from −150° at 1.7 GHz to 180°, as shown in Figure 12b. The magnitude and phase of the transmission coefficient are shown in Figure 13. The magnitude of |S_21_|is changed by 10 dB over the operating band due to an increase in tumor size, as shown in Figure 13a.

In order to monitor circularly polarized operations, the axial ratio is recorded, as shown in Figure 14, in both scenarios. The axial ratio is shifted down by 500 MHz due to the varying size of the tumor, as presented in Figure 14a. In the second scenario, with two sensors placed on the breast phantom, the operating band with an axial ratio of less than 3 dB is changed from 2.5 GHz to 3 GHz and from 5 to 8 GHz. With the presence of the tumor, the lower band of the CP band is shifted by 1 GHz and changed from 3.5 GHz to 4 GHz, as shown in Figure 14b.

Moreover, when the location of the tumor varies, the simulated results of the tumor shifting from the center (Z: offset distance from center presented in Figure 10) are also studied, as shown in Figure 15. Figure 15 shows the changes in the CPMS’s properties. When the tumor is both centralized and on the same axes between the sensors, the rate of change is elevated. However, the further the tumor from the antennas, the rate of properties decreases.

The magnitude of the reflection and transmission coefficient is changed by more than 30 dB when the antenna is placed far from the tumor by 30 mm. The axial ratio of the proposed microwave sensor also varied by changing the tumor location and size. Then the realized antenna gain is also studied in the free space and in the presence of a breast phantom, as shown in Figure 15d. The antenna acquires a gain of 6 dBi without the presence of a breast model and −7 dBi while placed on a breast model. This could be attributed to the dispersive, lossy properties of human tissues, so the antenna gain is reduced over the operating band by more than 13 dBi.

Simulated radiation patterns in the air and with breast models at different angles are presented in Figure 16 and Figure 17 with two main planes, phi = 0° and 90° and at 2.1 and 4.5 GHz, respectively.

The radiation pattern of the proposed sensor is an omni-radiation pattern, which is accepted in wireless communication.

## 4. Experimental Results

This section will present the fabrication and measurements of the proposed CPMS. Experimental results will be compared with simulations. In Section 4.1, Measurements in air and off-body will be presented. This includes measurements of magnitude and phase for reflection coefficients, transmission coefficients, and gain, as well as an axial ratio versus frequency. Moreover, measurements for the proposed sensor on breast phantoms are performed and analyzed. SAR levels are calculated and presented in Section 4.2.

### 4.1. Sensor Measurements

The proposed microwave sensor antenna is drawn to scale on Wilcon Embroidery Studio software for accurately produced antennas using a computer numerical control (CNC) milling machine and fabricated, as shown in Figure 18a. ShieldiT is used as a conductive fabric for the proposed antenna-based sensor. The proposed sensor is stitched to cotton fabric acting as substrate, as shown in Figure 18a. It is further measured using Rohde and Schwarz vector network analyzer with Model no. ZVA67 and bandwidth from 10 MHz up to 67 GHz. The reflection coefficient is recorded and compared with simulations in Figure 18b.

Despite being highly flexible and easily incorporated into clothing, textile antennas present a number of manufacturing issues. The operating band, as well as the impedance matching performance, alter due to the textile’s susceptibility to changes in humidity, temperature, as well as pressure. Humidity and pressure can affect the performance of flexible textile-based antennas. This includes bending and stretching effects, as well as washing cycles. This affects its resonance frequency and radiation characteristics, particularly when circular polarization is required [2]. Figure 18a shows the fabricated antenna while being tested for bending. The differences between simulated and measured results at low and high frequencies could be attributed to several reasons. First, the effects of the coaxial cable used in the measurement. Second, the conductor fabric and cotton substrate are uneven, and there is air between the layers. Third, soldering *the* SMA connector with silver paste to the textile feeder line. Fourth, other electromagnetic interference signals in the atmosphere and the ideal model used in simulations, as well as manufacturing and measurement tolerances in the positioned antenna. Moreover, the effect of bending on both *θ* and *ϕ* axis are also shown in Figure 18b,c. Figure 18b,c indicate that CPMS textile-based antenna maintains its performance whenever bending occurs in either direction (along θ or ϕ axes). Moreover, the response of the fabricated sensor is compared, as shown in Fig 19a and Figure 19b, at dry, wet, and after wash conditions. It shows stable and repeated performance and great potential to be implemented on clothes.

Finally, the proposed CPMS gain and radiation efficiency are measured in the free space in an anechoic chamber in the microwave lab at the Faculty of Engineering, Ain Shams University. The measurement system is composed of Near-field Systems, Inc. 700S-30 (CA, 90502, USA), with one wideband double-ridged horn antenna and NSI-RF-RGP-10, connected to a Vector Network Analyzer, Rohde & Schwarz ZVB14) (Columbia, MD, USA), as shown in Figure 20a. Figure 20b shows that the realized gain of the antenna in free space is around 4 dBi over the operating band. The efficiency is also calculated with an acceptable level of 60%. Efficiency is an important parameter to be recorded for microwave-based sensors in WBAN or sensing systems attached to human phantoms. Moreover, the antenna gain is also calculated at different bending angles along θ axes in Figure 20c. It can be deduced that the performance of CPMS has good purity in the operating band, but the gain has decreased. The curvature of the antenna decreases the antenna’s effective size and accordingly reduces gain. However, the performance is commendable, making the proposed antenna appropriate for wearable technology.

Different tumor models are shown in Figure 21; as the tumor size increased from 0 to 25 mm, the resonant frequency shifted down from 4.5 GHz to 4 GHz, as shown in Figure 21a, while the phase irregularly fluctuated from 180° to −180° at a different frequency with different bandwidth.

### 4.2. SAR Measurements

The specific absorption rate (SAR) can determine how much power these radiators, acting as sensors, are absorbed by human tissues. The antenna can be called safe if its maximum SAR value does not exceed 1.6 W/Kg based on the IEEE C95.3 standard. The simulated SAR is calculated using the CST simulator at 2 GHz and 5 GHz. SAR values at 100 mW that transmitted power of 2.32 and 0.98 W/Kg, respectively, were recorded for 1 g and 10 g of tissue standards, respectively. For further lower transmitted power of 50 mW, SAR levels of 1.163 W/Kg and 0.489 W/Kg power are recorded for 1 g and 10 g. This falls in the safe zones for SAR of 1.6 W/Kg and 2 W/Kg for 1 g and 10 g, respectively. Moreover, The SAR level for the proposed sensor is measured in SAR Lab at the Electronics Research Institute, as shown in Figure 22. The measured SAR levels are recorded in Table 2 at different power levels of 5, 10, 15, and 20 dBm. SAR values for the proposed sensors maintained safe levels at both 1 g and 10 g standards.

## 5. Conclusions

Various research groups have focused on developing microwave breast imaging techniques that are non-invasive, harmless to women, and suitable for regular screening. Microwave-based breast imaging techniques rely on the contrast in electrical properties of normal and malignant breast tissues. UWB antennas are more attractive in tumor detection applications as they match both the medical band and the ISM band, requiring less power and improved safety levels. In this work, a circularly polarized, flexible, conformal, wideband, CPW-based monopole antenna in the band from 1.8 to 8 GHz is presented, fabricated, and measured.

The proposed antenna is fully composed of textiles for both conductor and substrate parts with an overall size of 33.5 × 33.5 mm^2^ and an average gain of 6 dBi. The antenna as a single element and two elements are simulated in air and in proximity to breast and tumor models. Reflection and transmission coefficients results show that the presence of cancerous cells can induce a shift in resonant frequency by 0.5 GHz and a change in magnitude by 10 dB and a shift of 1 GHz to the axial ratio band. Safety SAR levels below 1.6 W/kg could be maintained with transmitted power of 50 mW. A 3D-breast phantom is modeled in simulation software and fabricated of physical phantoms for both breast and tumor and applying measurements with it. The proposed work in this paper demonstrates the development of fully textile antennas as sensors for Breast Cancer Detection.

The proposed monopole antenna is compared with some of the most recently published works for biomedical applications in [2,3,4,5,6,7,40,41,42,43,44,45,46,47], as shown in Table 3. The total performance of an antenna is determined by several factors, including fractional bandwidth, average gain, radiation efficiency, physical dimensions, etc. The figure of merit presented in [47] is used to calculate the overall performance of the designed planar antenna. It is recorded in [47] that increasing the figure of merit indicates an antenna’s overall performance enhancement. The work presented in [47] has a figure of merit value of −37 dB, while our proposed antenna achieved −20 dB.

Table 3 shows that most published textile antennas are linearly polarized [3,40,41,42,43,46,47] with limited bandwidth operation. For SAR calculations, most published textile-based sensors acquire a low SAR at low transmitted power levels. The proposed sensor acquires a low SAR of 0.489 W/Kg at 50 mW. The proposed CPMS acquires a higher gain of 6 dBi wider bandwidth and an axial ratio percentage of 73% compared with other CP antennas reported in the literature [2,5,6,7,45]. The proposed textile microwave sensor acquires a wide bandwidth, moderate gain, and circularly polarized output with a small footprint and safe SAR values. All these aspects make the proposed sensor a good candidate for use in wireless body area networks (WBANs) and breast cancer imaging. Multiple copies of this sensor could be integrated within women’s bras to monitor breasts in the comfort of their homes. From previous research, it could be indicated that the proposed CPMS is very suitable for breast cancer detection and wearable garments, such as a Smart Bra for different breast cancer detection and biomedical applications.

## Figures and Tables

**Figure 1 micromachines-14-00586-f001:**
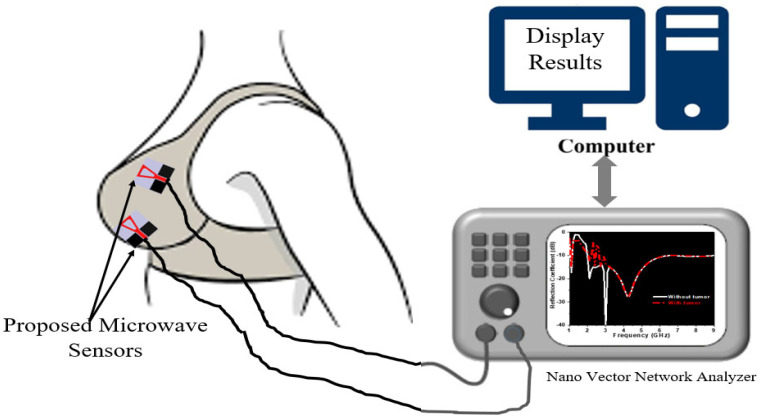
Proposed Smart Bra for breast cancer monitoring detection.

**Figure 2 micromachines-14-00586-f002:**
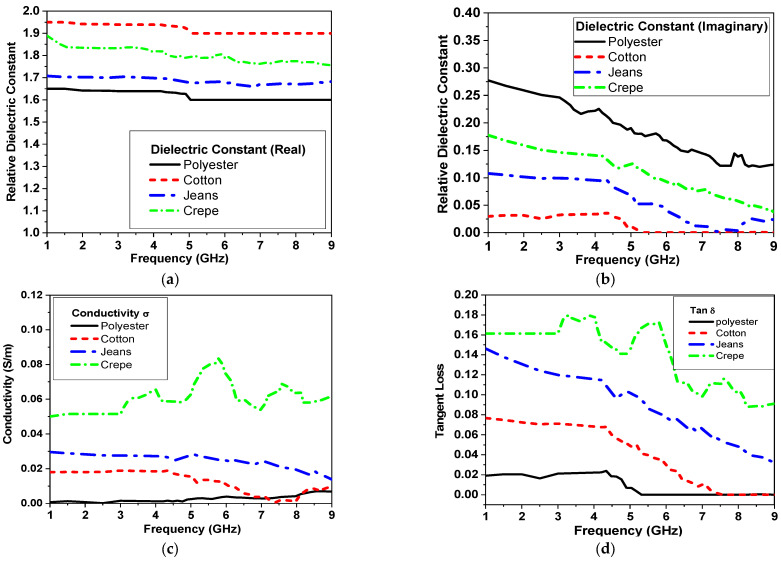
Measured electrical properties of different textiles versus frequency: (**a**) real part of dielectric constant (ε′), (**b**) imaginary part of dielectric constant (ε″), (**c**) Conductivity (σ) and (**d**) loss tangent (tanδ).

**Figure 3 micromachines-14-00586-f003:**
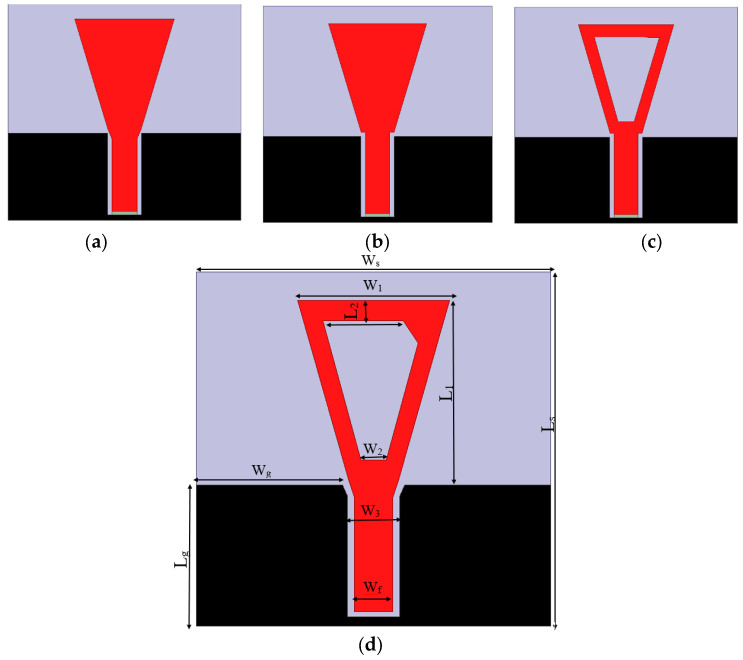
The design steps of CPW-CP monopole antenna, (**a**) trapezoidal monopole and (**b**) trapezoidal monopole with notches, (**c**) trapezoidal monopole after etching trapezoidal slot and (**d**) Configuration of the circularly polarized CPW-monopole antenna.

**Figure 4 micromachines-14-00586-f004:**
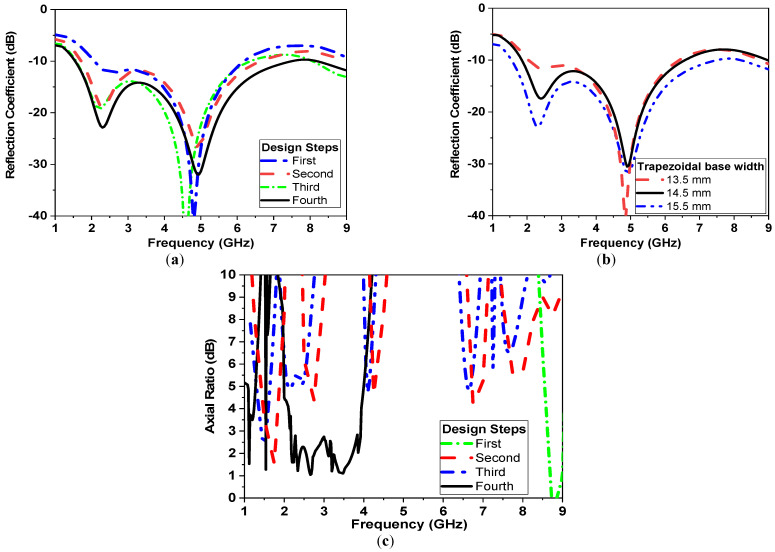
(**a**) |S_11_| versus frequency at different design steps, (**b**) |S_11_| versus frequency at different trapezoidal base widths, and (**c**) axial ratio (AR) in dB versus frequency at different design steps.

**Figure 5 micromachines-14-00586-f005:**
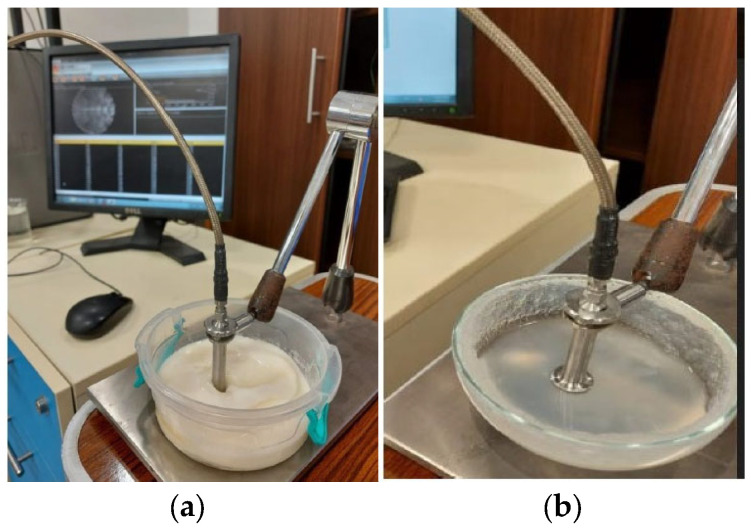
Characterization of fabricated phantoms using DAK dielectric probe station (**a**) breast and (**b**) tumor.

**Figure 6 micromachines-14-00586-f006:**
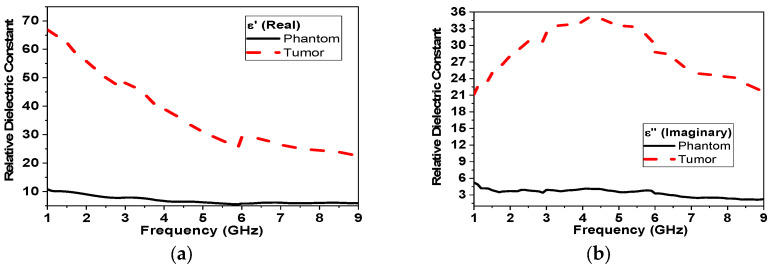
Measured electrical properties of breast phantom and tumor models versus frequency: (**a**) real part of dielectric constant (ε′), (**b**) imaginary part of dielectric constant (ε″), (**c**) conductivity (σ), and (**d**) imaginary part of conductivity (σ).

**Figure 7 micromachines-14-00586-f007:**
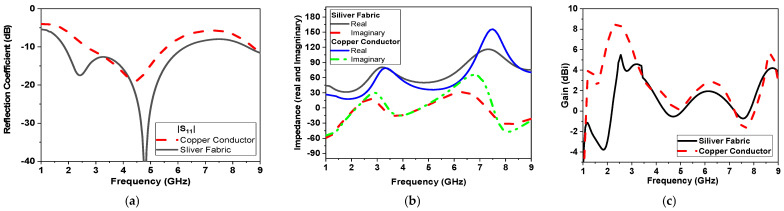
(**a**) Simulated reflection coefficient in dB, (**b**) simulated impedance (real and imaginary), and (**c**) simulated gain in dBi versus frequency for the two prototypes using copper clad and silver fabric.

**Figure 8 micromachines-14-00586-f008:**
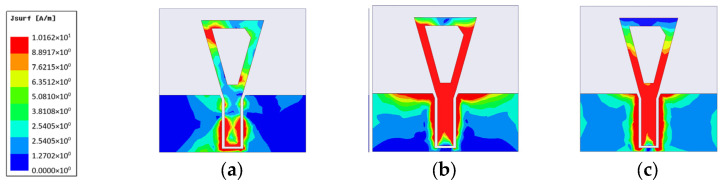
Surface current distribution of the proposed CPMS at different operation frequencies (**a**) 2.5 GHz, (**b**) 3.5 GHz, and (**c**) 5 GHz.

**Figure 9 micromachines-14-00586-f009:**
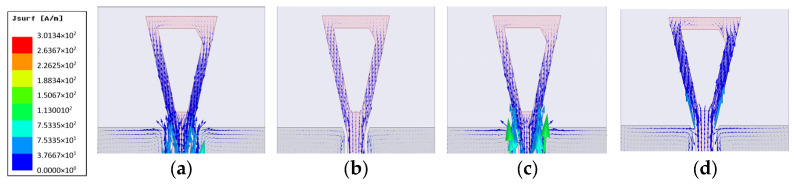
Surface current vector distribution of the proposed CPMS within CP frequency band (**a**) 2.2 GHz, (**b**) 2.4 GHz, (**c**) 2.5 GHz, and (**d**) 3.5 GHz.

**Figure 10 micromachines-14-00586-f010:**
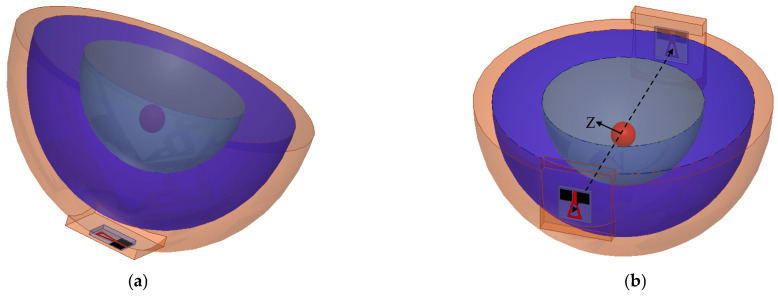
Breast with tumor models scanned using (**a**) one antenna and (**b**) two antennas.

**Figure 11 micromachines-14-00586-f011:**
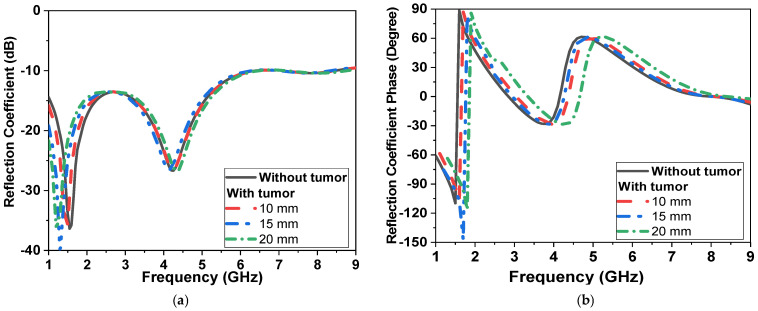
Simulation |S_11_| of the proposed CPMS sensor with tumor model (first scenario shown in Figure 10a) (**a**) magnitude in dB, (**b**) phase in degrees.

**Figure 12 micromachines-14-00586-f012:**
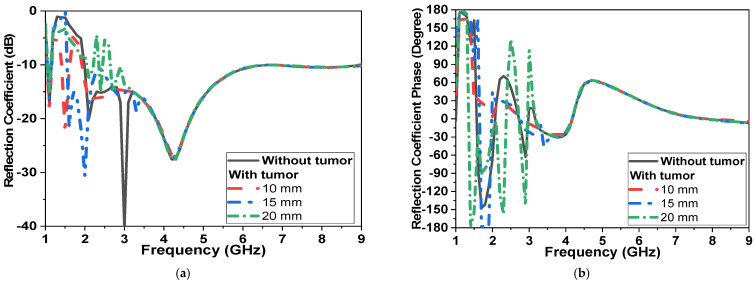
Simulated |S_11_ | of the proposed CPMS sensor with tumor model (second scenario shown in Figure 10b) (**a**) Magnitude in dB, (**b**) Phase in degrees.

**Figure 13 micromachines-14-00586-f013:**
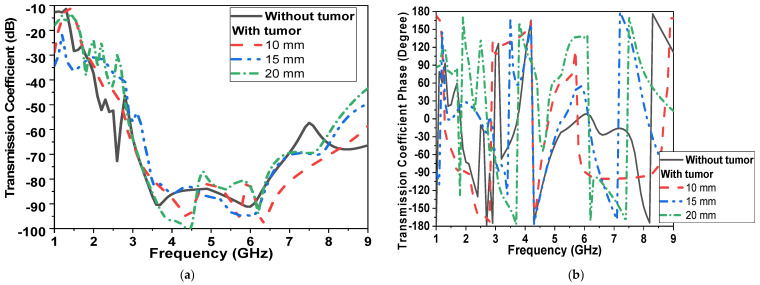
Simulated |S_21_| of proposed CPMS with tumor model (second scenario shown in Figure 10b) (**a**) Magnitude in dB, (**b**) Phase in degrees.

**Figure 14 micromachines-14-00586-f014:**
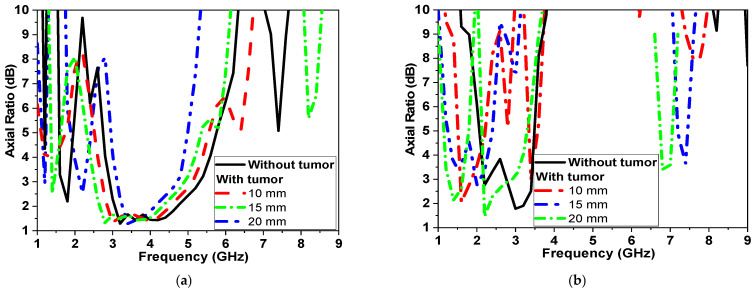
Simulated AR of proposed CPMS at different tumor sizes (**a**) First scenario shown in Figure 10a; (**b**) second scenario shown in Figure 10b.

**Figure 15 micromachines-14-00586-f015:**
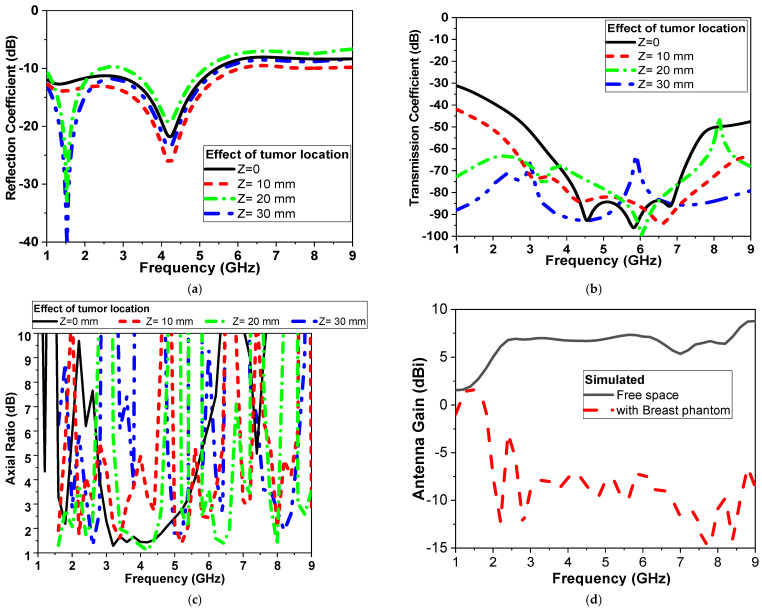
Simulated of proposed CPMS parameters with tumor location (second scenario shown in Figure 10b) (**a**) |S_11_| (**b**) |S_21_| in dB, (**c**) Axial ratio and (**d**) antenna Gain.

**Figure 16 micromachines-14-00586-f016:**
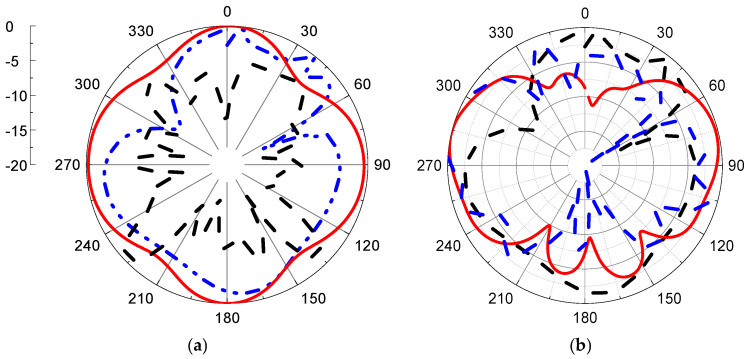
Radiation pattern for proposed antenna with and without breast model (**a**) φ = 0°, (**b**) φ = 90°. Red line: simulated without phantom, black dashed line: simulated with phantom, and blue dash\-dot line: measured without phantom at 2.1 GHz.

**Figure 17 micromachines-14-00586-f017:**
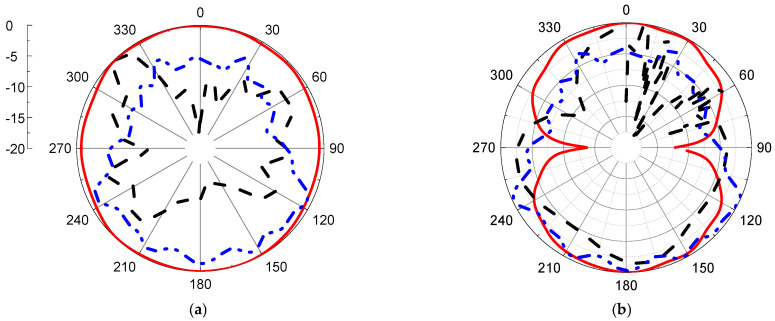
Radiation pattern for proposed antenna with and without breast model (**a**) φ = 0°, (**b**) φ = 90° Red line: simulated without phantom, black dash line: simulated with phantom, and blue dash-dot line: measured without phantom) at 4.5 GHz.

**Figure 18 micromachines-14-00586-f018:**
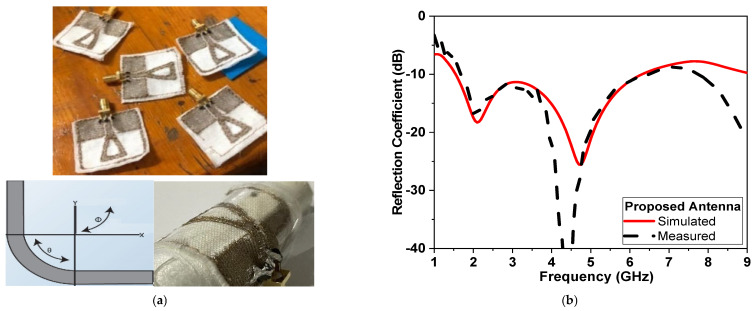
(**a**) The fabricated textile-based CPMS sensor, (**b**) Comparison of the CPMS sensor reflection coefficient between measured and simulated (**c**) Measured reflection coefficient of the CPMS sensor bending effect on *θ*^o^ and (**d**) Reflection coefficient of the CPMS sensor bending effect on *ϕ*^o^.

**Figure 19 micromachines-14-00586-f019:**
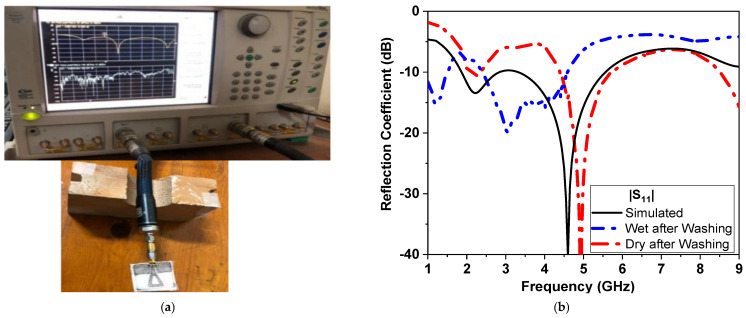
(**a**) Reflection measurement setup for fabricated CPMS sensor connected to vector network analyzer (VNA) and (**b**) simulation and measurement of reflection coefficient in dB versus frequency at different washing stages.

**Figure 20 micromachines-14-00586-f020:**
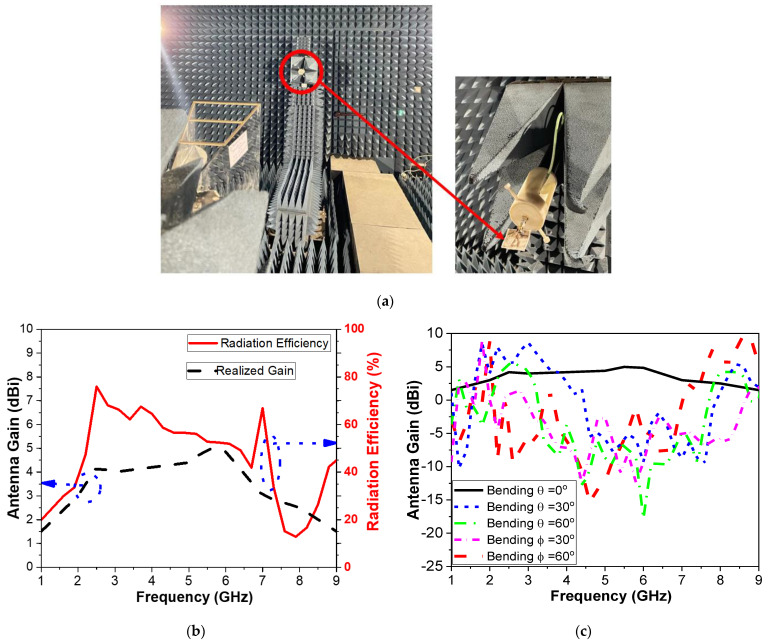
(**a**) Photo of the gain setup, (**b**) gain and radiation efficiency of the CPMS, and (**c**) effect of gain bending on θ axes, as in Figure 18a.

**Figure 21 micromachines-14-00586-f021:**
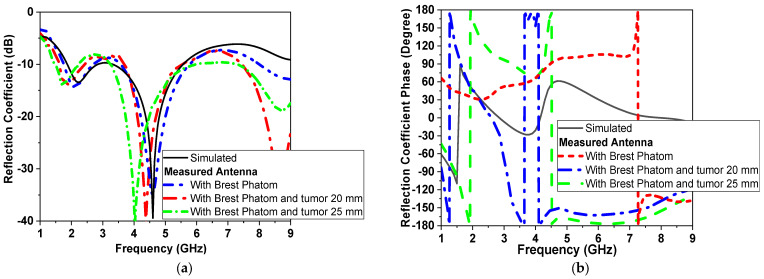
Measured |S_11_| of sensor with breast and tumor models (**a**) magnitude and (**b**) phase.

**Figure 22 micromachines-14-00586-f022:**
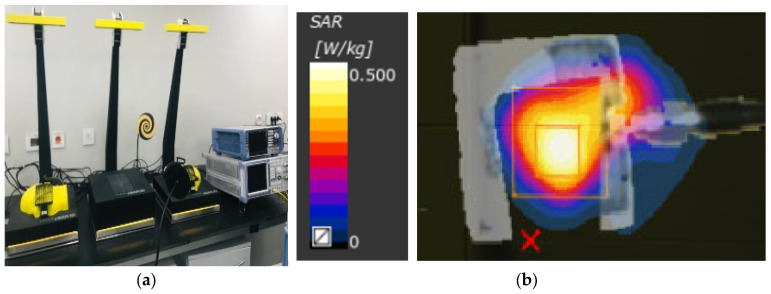
SAR measurements of the proposed antenna-based sensor (**a**) measurement setup and (**b**) SAR distribution of the CPMS.

**Table 1 micromachines-14-00586-t001:** The optimized dimensions of the proposed antenna (all dimensions in mm).

*W_g_*	*L_g_*	*W_f_*	*W* _3_	*W* _2_	*W* _1_	*L* _2_	*L* _1_	*W_s_*	*L_s_*
14.5	13.5	4	5	2.5	14.5	2	17.5	33.5	33.5

**Table 2 micromachines-14-00586-t002:** Comparison of measured SAR levels for the circularly polarized monopole sensor.

SAR Level for CPMS		Power Level
10 g	1 g	(dBm)
0.07	0.201	5
0.086	0.293	10
0.211	0.594	15
0.333	0.944	20

**Table 3 micromachines-14-00586-t003:** Performance Comparison between proposed antenna sensor and previously reported architectures.

BW of AR (GHz) & %	CP/LP	SAR (W/Kg) in 10 g	Fabrications	Dimensions (W × L × h) mm^3^	Gain (dBi)	BW (GHz)	Ref.
2.39–2.57 (7%)	CP	81.5 at (1 W)	Rogers 3010	9.2 × 9.2 × 1.27	NM	2.39–2.57	[2]
-	LP(MIMO)	1.93	Denim and felt	24 × 24 × 1.4	5.72	4.8–30	[3]
5.2–7.1	CP	367.86 mW at (2 W)	Felt	32:5 × 42 × 1	5.7	3.6–13	[5]
−31%							
2.3–2.864 (22%)	CP	NM	Felt	76 × 76× 1.5	4.9	1.086	[6]
5.730–5.955	CP	0.294	Flexible Panasonic R-F770	35 × 35 × 2.24	7.2	5.67–6.05	[7]
−4%							
-	LP	NM	Cordura	80 × 30 × 0.5	2.01	0.9, 1.8	[40]
-	LP	0.09	Denim	60 × 50 × 0.7	10.5	7–28	[41]
-	LP	NM	FR4	27 × 27 × 1.6	8.65	1.23–20	[42]
-	LP	3.28 × 10^−6^	Felt	36 × 18 × 3	7.81	3.1–6.5	[43]
-	Dual Polarized	NM	Kapton Polymidie	20 × 20 × 0.05	NM	2–4	[44]
2.38–2.51(5%)	CP	242	RT Duroid	10 × 10 × 0.3	−7.79	2.3–2.64	[45]
-	LP	0.0014 at (1 mW)	Polyster fabric	70 × 50 × 0.5	2.9	1.198–4	[46]
-	LP	1 (0.5 W)	Flexible Kapton Polyimide	13 × 13 × 0.125	4.4	8.6–14	[47]
1.8–4 (73%)	CP	0.489 (50 mW)	Cotton	33.5 × 33.5 × 2	6	1.8–8	This work

NM: (not mention in the paper), *CP*: circular polarized, *LP*: linear polarized.

## Data Availability

Not applicable.

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
