# Peer review of "Circularly Polarized Textile Sensors for Microwave-Based Smart Bra Monitoring System"

_micromachines, 2023, doi:10.3390/mi14030586_

Round 1

Reviewer 1 Report

In the manuscript micromachines-2181361, the authors presented a circularly polarized microwave sensor of 50 × 50 mm2 as smart bra for breast cancer detection, which worked in the bandwidth from 1.8 GHz to 8 GHz at |S11| ≤-10 dB and circularly polarized with AR ≤ 3 dB operation band from 1.8 to 4 GHz. The proposed circularly polarized sensor is continuation to the  developed SMART BRA in reference [23] by the authors. However, the present status of manuscript can not meet the stadard of Micromachines, so I recommed it to be revised by the following comments.
1. The preparation process for circularly polarized textile sensors needs to be briefly mentioned.
2. The performance of the circularly polarized textile sensor in the bending state needs to be tested and whether there are adverse effects needs to be discussed.
3. In the introduction, the innovation of  this research needs to be further clarified.
4. Comparing the reference 23 in the performance, the advantage of circularly polarizetion can be highlighted.
5. For wearable application, the repeated test of circularly polarized textile sensors may be carried out.
6. There are many typoes, including the name of the authors, so extensive editing of English language and style is required.

Author Response

Response Letter

Manuscript ID: micromachines-2181361

Manuscript Title: Circularly Polarized Textile Sensors for Microwave-based Smart Bra
Monitoring System

Authors: * Dalia N. Elsheakh *, Yasmine K. Elgendy, Mennatullah E. Elsayed,
Angie R. Eldamak

First, we would like to sincerely thank the editor and the reviewers for their valuable comments. We have considered all the reviewers’ comments and suggestions and have modified our manuscript accordingly.

We have also prepared detailed responses for reviewers’ comments. In the revised paper, we use highlighted yellow text for modified sections, figures & tables in the paper based on the reviewers’ comments.

Sincerely yours,

Comments of Reviewer 1

The authors presented a circularly polarized microwave sensor of 50 × 50 mm2 as smart bra for breast cancer detection, which worked in the bandwidth from 1.8 GHz to 8 GHz at |S11| ≤-10 dB and circularly polarized with AR ≤ 3 dB operation band from 1.8 to 4 GHz. The proposed circularly polarized sensor is continuation to the developed SMART BRA in reference [23] by the authors. However, the present status of manuscript cannot meet the standard of Micromachines, so I recommend it to be revised by the following comments.

Comments 1: The preparation process for circularly polarized textile sensors needs to be briefly mentioned.

  • Response:
  1. The design steps for the proposed sensors has been addressed in the manuscript in Sec 2.1, Fig.2 and Fig 3 in the updated manuscript: “The proposed textile-based sensor in the form of circularly polarized antenna operates in the band from 1.8 GHz to 8 GHz. The 33.5 × 33.5 mm2 sensor is composed of monopole antenna fed by coplanar waveguide (CPW) and placed on cotton substrate. Printed CPW-fed antenna structures have several desirable characteristics, including simplicity, low profile, small size, good radiation characteristics, and low radiation loss. Additionally, using antennas with CPW-fed structures helps wireless communication devices get smaller because their uniplanar structure makes it easier in integration with RF/microwave circuits and edge-fed connector boards [1].The proposed antenna has a trapezoidal radiator patch with dimensions of W1 and W2, respectively and height L1 as shown in Table 1 and Fig.3. The The proposed design is developed to the final structure through three steps shown in Fig.2. First, the design starts with trapezoidal monopole radiator with width W1  of 14.5 mm and and W2 of 3 mm, respectively as shown in Fig. 2 (a). The antenna has a 50 Ω transmission feeding CPW line of width (Wf) 4 mm and  5 mm air gaps shown in Fig.3. The proposed design is developed to the final structure shown in Fig.3.The reflection coefficient response against frequency is shown in Fig. 4(a). The antenna acquires a bandwidth from 2.3 to 5.5 GHz at |S11| ≤ -10 dB. The length of the large base of trapezoidal is optimized to achieve wider bandwidth as shown in Fig. 4(b). Second, two notches are introduced at the the transmission feeding line as shown in Fig. 2(b) to improve the impedance matching. This modification extends operating bandwidth  to 6 GHz as shown in Fig. 4(a). Third, a trapezoidal shaped slot is craved with dimensions 10 mm and 2.5 mm, respectively as shown in Fig. 2(c). This further extends bandwidth to start from 1.8 GHz to 6 GHz as shown in Fig. 4(a). Finally, the symmetry of the trapezoidal radiator is modified by adding a trapezoidal corner shown in Fig. 3. This help to induce circularly polarized operation and is validated by plotting axial ratio (AR) shown in Fig. 4(c).”
  • (b)                                                      (c)

Figure 2. The design steps of CPW-CP monopole antenna, (a) trapezoidal monopole and (b) trapezoidal monopole with notches and (c) trapezoidal monopole trapezoidal slot etched.

Figure 3. Configuration of the circular polarized CPW-monopole antenna.

  1. In terms of fabrication and preparation process, it is presented in Sec 4.1 in the updated manuscript as follows: “The proposed microwave sensor antenna is drawn to scale on Wilcon Embroidery Studio software for accurately produced antennas using computer numerical control (CNC) milling machine and fabricated as shown in Fig.18(a). ShieldiT is used  as a conductive fabric for the proposed antenna-based sensor. The proposed sensor is stitched to cotton fabric acting as substrate as shown in Fig. 18(a). It is further measured using Rohde and Schwarz vector network analyzer with Model no. ZVA67 and bandwidth from 10 MHz up to 67 GHz. Reflection coefficient is recorded and compared to simulations in Fig.18(b).”
  2. Cicrulary polarized performance is also monitored by plotting current vectors. This is updated in the revised manuscript in Sec 3.1 and in Fig. 9 as follows: “Morover, the current vector is also plotted in Fig.9. This helps in monitoring orthognal current components at specific operating bands and persuing circular polarized operation as shown in Fig. 9.”

(a)

(b)

(c)

(d)

Figure 9. Surface current vector distribution of the proposed CPMS within CP frequency band (a) 2.2 GHz, (b) 2.4 GHz, (c) 2.5 GHz and (d) 3.5 GHz.

Comments 2: The performance of the circularly polarized textile sensor in the bending state needs to be tested and whether there are adverse effects needs to be discussed.

Response: Thank you for your helpful comments. The circularly polarized textile sensor in the bending effect is simulated on the breast phantom by using 3D electromagnetic simulator Computer Simulation Technology (CST) as shown in following figure. Then the results is added as following:

(a)

(b)

(c)

(d)

Figure X. SAR calculation by using CST simulator at different frequency (a) 1.8 GHz, (b) 2.1 GHz, (c) 2.45 GHz and (d) 5.2 GHz.

  1. The effect of bending on the performance of the circularly polarized textile sensors has been added to the updated manuscript in Sec 4.1 and Fig. 18 as follows: “Several factors can affect the performance of flexible textile-based antennas. This includes bending and stretching effects as well as washing cycles. This affects its resonance frequency and the radiation characteristics particularly when circular polarization is required [2]. Fig. 18 (a) shows the fabricated antenna while being tested for bending. The effect of bending on both q and f axis are also shown in Fig. 18(b) and (c). Fig.18(b) and (c) indicate that CPMS textile-based antenna maintains its performance whenever bending occurs in either direction (along q or f axes).”

(a)

      (b)

(c)

Figure 18. (a) The fabricated textile-based CPMS sensor, (b) Measured reflection coefficient of the CPMS sensor bending effect on qo and (c) Reflection coefficient of the CPMS sensor bending effect on fo.

  1. In Sec 4.1, also the gain has been calculated at different bending angles as shown in Sec 4.1 and Fig. 20 as follows: “Finally, the proposed antenna gain is calculated in both cases with and without breast model phantom as shown in Fig. 20 (a). The antenna acquires gain of 6 dBi without presence of breast model and -7dBi while placed on Breast model. This could be attributed to the dispersive, lossy properties of human tissues. The antenna gain is also calculated at different bending angles along q axes in Fig.20(b). It can be deduced that the performance of CPMS has good purity in the operating band, but the gain has decreased. The curvature of the antenna decreases the antenna's effective size and accordingly reduces gain. However, the performance is commendable, making the proposed antenna appropriate for wearable technology.”

  • (b)

Figure 20.  (a) Configuration of the CPMS and (b) Effect of bending on q and f axes.

  1. The effect of bending on the antenna gain is shown in following figure by using 3D electromagnetic simulator Computer Simulation Technology (CST)

Figure XX. The gain versus frequency of the proposed sensor

Comments 3: In the introduction, the innovation of this research needs to be further clarified.

Response: The innovation of this research has been revised and highlighted in more than position in the manuscript.

  1. In the Abstract as follows: “This paper presents a conformal and biodegradable circularly polarized microwave sensor (CPMS) that can be utilized in several medical applications. The proposed textile sensor can be implemented in Smart Bra system for breast cancer detection (BCD) and wireless body area network (WBAN). The proposed sensor is composed of a wideband circularly polarized (CP) textile-based monopole antenna with an overall size of 33.5 × 33.5 mm2 (0.2 lo × 0.2 lo) and CPW feed line. The radiating element and ground are fabricated using silver conductive fabric and stitched to cotton substrate of thickness 2 mm. In the proposed design, slot is etched in the radiating element to extend bandwidth from 1.8 to 8 GHz at |S11| ≤-10 dB. It realizes a circularly polarized output with AR ≤ 3 dB operation band from 1.8 to 4 GHz and average gain 6 dBi. The proposed CPMS’s performance is studied in both off-body (air) and on-body in proximity to breast models with and without tumor using near-field microwave imaging. Moreover, the axial ratio is recorded as a feature for a circularly polarized antenna and added another degree of freedom for cancer detection and data analysis. It assists in detecting tumor in the breast by analyzing the magnitude of the electrical field components in vertical and horizontal direction. Finally, the radiation properties are recorded as well as specific absorption rate (SAR) to ensure safety operation.  The proposed CPMS covers a bandwidth of 1.8–8 GHz with SAR values following the 1 g and 10 g standards. The proposed work demonstrates the feasibility to use textile antennas in wearables, microwave sensing systems and wireless body area networks (WBANs).”
  2. In the Introduction:, “In this paper, we introduce an ultra-wideband (UWB) antenna that can be utilized as sensors in two major applications: (1) wireless body area networks (WBANs) and (2) breast cancer imaging. This paper presents a fully textile CPW monopole antenna with operating bandwidth from 1.8 GHz to 8 GHz and circularly polarized output in the band of 1.8 - 4 GHz.”
  3. In the Introduction: “The proposed circularly polarized sensor is continuation to the work in [30] to develop Smart Bra as shown in Fig. 1. In [30], the textile-based sensor is presented as linearly polarized microstrip antenna and has an impedance bandwidth from 1.8 to 2.4 GHz and from 4 up to 10 GHz at |S11| ≤ −10 dB. By enabling circularly polarized output, a new monitoring indicator, axial ratio, is added. Axial ratio will be recorded in addition to S-parameters (|S11 | for reflection and |S21 | for transmission). The proposed CPMS has a broad axial ratio (AR) bandwidth that allows efficient operation in challenging conditions and in proximity to lossy, dispersive, and heterogeneous human tissues. Moreover, it will be also used in analyzing data measured from breast. This will increase the samples of collected data and will contribute to increasing the detection accuracy. The proposed system is meant to help women in early detection and continuous monitoring for breast in the comfort of their homes.”
  4. In conclusion: “Table 3 shows that most of published textile antennas are linear polarized [3, 40-43, 46, 47] with limited bandwidth operation. For SAR calculations, most of published textile-based sensor acquire low SAR at low transmitted power levels. The proposed sensor acquires low SAR of 0.489 W/Kg at 50 mW. The proposed CPMS acquire higher gain of 6 dB wider bandwidth and axial ratio percentage of 73% compared to another CP antennas reported in literature [2, 5-7, 45]. From the previous it could be indicated that the proposed CPMS is very suitable for breast cancer detection and wearable garments as smart bra for different breast cancer detection and biomedical applications.”

Comments 4: Comparing the reference 23 in the performance, the advantage of circularly polarization can be highlighted.

Response:

  1. By comparing the monopole in Ref [23 or 30 in the revised manuscript] and the one in the updated manuscript, both designs are flexible, textile-based, wearable. In [30], the textile-based sensor is presented as linearly polarized microstrip antenna and has an impedance bandwidth from 1.8 to 2.4 GHz and from 4 up to 10 GHz at |S11| ≤ −10 dB. In the submitted manuscript, the sensor has an impedance bandwidth from 1.8-6 GHz and higher gain of 6 dBi compared to 3.5 dBi for the one published in [30]. The footprint of the proposed sensor in the submitted manuscript is slightly 33.5 × 33.5 × 2 mm compared to 24 × 45 × 2 mm. In [30], the textile-based sensor is presented as linearly polarized microstrip antenna. By enabling circularly polarized output, a new monitoring indicator, axial ratio, is added. Axial ratio will be recorded in addition to S-parameters (|S11 | for reflection and |S21 | for transmission). The proposed CPMS has a broad axial ratio (AR) bandwidth that allows efficient operation in challenging conditions and in proximity to lossy, dispersive, and heterogeneous human tissues. Moreover, it will be also used in analyzing data measured from breast. This will increase the samples of collected data and will contribute to increasing the detection accuracy. The proposed system is meant to help women in early detection and continuous monitoring for breast in the comfort of their homes.”

  1. The advantage of circularly polarization has been highlighted in Introduction as follows: “Wearable antennas can bend and stretch while in use and accordingly the radiation properties are varied. Thus, CP antenna is favored in developing flexible wearable sensors and systems. CPMS microwave sensor waves contain both vertical and horizontal electric field components. Thus, receiver antennas can pick up radiation waves from a variety of angles; when these waves are reflected or refracted. Moreover, circular polarization (CP) antennas reduce polarization mismatch, flexibility in terms of orientation and increased mobility. Moreover, it is a good choice for encountering multipath fading and establishing reliable channels when compared to linearly polarized (LP) antennas, especially in crowded interior environments [1-5]. However, only a few works focus on the CP MS wearable antennas [1-3, 5-7].

Several Circularly polarized (CP) antennas have been studied, including microstrip patch antennas [31], filtering antennas, and anisotropic artificial ground plane loaded antennas [32]. The truncated corner of the microstrip patch antenna in [33] are intended to produce CP unidirectional radiation. However, the coaxial probes that feed these antennas make them difficult to use in wearable applications. A wearable biotelemetric system with a tiny CP antenna integrated with band-pass filter circuits is proposed in [31]. However, it uses a rigid substrate that cannot conform to the human body.  In [31], a flexible CP antenna operating at industrial, scientific, and medical (ISM) band 2.4 GHz is investigated. However, these wearable CP antennas have a limited axial ratio (AR) bandwidth, which could cause performance degradation due to a potential frequency offset”

  1. The advantage of circularly polarization has been highlighted in Introduction as follows: The proposed circularly polarized sensor is continuation to the work in [30] to develop Smart Bra as shown in Fig. 1. In [30], the textile-based sensor is presented as linearly polarized microstrip antenna and has an impedance bandwidth from 1.8 to 2.4 GHz and from 4 up to 10 GHz at |S11| ≤ −10 dB. By enabling circularly polarized output, a new monitoring indicator, axial ratio, is added. Axial ratio will be recorded in addition to S-parameters (|S11| for reflection and |S21| for transmission). The proposed CPMS has a broad axial ratio (AR) bandwidth that allows efficient operation in challenging conditions and in proximity to lossy, dispersive, and heterogeneous human tissues. Moreover, it will be also used in analyzing data measured from breast. This will increase the samples of collected data and will contribute to increasing the detection accuracy. The proposed system is meant to help women in early detection and continuous monitoring for breast in the comfort of their homes.”

Comments 5:  For wearable application, the repeated test of circularly polarized textile sensors may be carried out.

Response: Repeated test of the proposed CPMS has been conducted. This has been validated by measuring the reflection coefficient values versus frequency for three copies of the same design. The measurements are shown in the following figure. Most of these copies of the CPMS sensor operated in the required band. The difference of these results could be attributed from the soldering of sliver paste. Additionally, compared to traditional printed circuit board (PCB) processes, textile-based systems' manufacturing tolerances are much looser, resulting in a significantly greater range of uncertainty.

Figure XXX.  The reflection coefficient values versus frequency for three copies of the proposed microwave sensors.

Comments 6:  There are many typoes, including the name of the authors, so extensive editing of English language and style is required.

Response: The whole manuscript has been reviewed in-terms of typos, English language and style. All dimensions, values, figures and tables are revised as well.

Reviewer 2 Report

The authors presented Circularly Polarized Textile Sensors for Microwave-based Smart Bra Monitoring System. The concept is exciting, and the simulation results are reasonably good, showing strong reconfigurability. I have the following suggestions before accepting it for publication:

- In the introduction part, the author must discuss the techniques for circular polarised printed antenna systems [1] and discuss breast cancer detection for body area network applications [2].

[1] Broadband Circular Polarised Printed Antennas for Indoor Wireless Communication Systems: A Comprehensive Review. Micromachines 202213, 1048. https://doi.org/10.3390/mi13071048.

[2] Full Ground Ultra-Wideband Wearable Textile Antenna for Breast Cancer and Wireless Body Area Network Applications. Micromachines 202112, 322. https://doi.org/10.3390/mi12030322.

- Since the antenna is applied for Microwave imaging applications, I want to see the Specific Absorption Rate (SAR) calculation weights in terms of 1 gram up to 10 grams of tissue?

 -It needs to be evident how the proposed design can be used to address the research gaps of the current study. Significant improvements need to be made by the authors to emphasize the significance of this paper. For example, how many percentages of Axial Ratio improvements are obtained by the CP proposed design?

- What impact does the monopole antenna in the proposed wearable antenna have on the circularly polarized technique at 1.8 GHz to 8 GHz?

- The quality of the figures presented in this paper could be better, and the fonts need to be more significant to read. Please improve the quality of the figures, especially for Figure 12.

- The differences between the simulated results and the measured results in Figure 21 should be discussed more.

- References in Table 3 are incorrect; please correct them!

- The Conclusions should be rewritten with the updated results above.

The authors are required to revise the comments mentioned above carefully.

Author Response

Response Letter

Manuscript ID: micromachines-2181361

Manuscript Title: Circularly Polarized Textile Sensors for Microwave-based Smart Bra
Monitoring System

Authors: * Dalia N. Elsheakh *, Yasmine K. Elgendy, Mennatullah E. Elsayed,
Angie R. Eldamak

First, we would like to sincerely thank the editor and the reviewers for their valuable comments. We have considered all the reviewers’ comments and suggestions and have modified our manuscript accordingly.

We have also prepared detailed responses for reviewers’ comments. In the revised paper, we use highlighted yellow text for modified sections, figures & tables in the paper based on the reviewers’ comments.

Sincerely yours,

Comments of Reviewer 2

The authors presented Circularly Polarized Textile Sensors for Microwave-based Smart Bra Monitoring System. The concept is exciting, and the simulation results are reasonably good, showing strong reconfigurability. I have the following suggestions before accepting it for publication:

Comments 1: In the introduction part, the author must discuss the techniques for circular polarised printed antenna systems [1] and discuss breast cancer detection for body area network applications [2].

[1] Broadband Circular Polarised Printed Antennas for Indoor Wireless Communication Systems: A Comprehensive Review. Micromachines 202213, 1048. https://doi.org/10.3390/mi13071048.

[2] Full Ground Ultra-Wideband Wearable Textile Antenna for Breast Cancer and Wireless Body Area Network Applications. Micromachines 202112, 322. https://doi.org/10.3390/mi12030322.

Response: Thank you very much for your effort in reviewing our manuscript. As per the received feedback, the techniques for circular polarized printed antenna systems and breast cancer detection for body area network have been added to the revised manuscript as follows in:

  1. In the Introduction, “A wireless body area network (WBAN) is a group of low-power, miniature, light-weight wireless sensors that monitor internal and external body health processes. WBANs help in monitoring physiological behavior patterns in humans and studying disease prevention and control [1]. The WBAN's ultra-lightweight, wearable sensors can be either off-body or on-body [2] and could be operated in MIMO systems [3-5]. Numerous applications of WBAN have embraced several devices on a large-scale including brain recording, glucose monitoring, intracranial pressure monitoring as well as breast cancer detection [2, 6]. WBAN use wearable, light weight antennas as a feature to transmit and receive signals to the human body. In this paper a wearable bra is proposed to monitor and scan breast cancer. This Smart Bra is presented as a component of a WBAN system such that women won't need to visit the hospital regularly [7].
  2. In the Introduction, The advantage of circularly polarization has been highlighted in Introduction as follows: “Wearable antennas can bend and stretch while in use and accordingly the radiation properties are varied. Thus, CP antenna is favored in developing flexible wearable sensors and systems. CPMS microwave sensor waves contain both vertical and horizontal electric field components. Thus, receiver antennas can pick up radiation waves from a variety of angles; when these waves are reflected or refracted. Moreover, circular polarization (CP) antennas reduce polarization mismatch, flexibility in terms of orientation and increased mobility. Moreover, it is a good choice for encountering multipath fading and establishing reliable channels when compared to linearly polarized (LP) antennas, especially in crowded interior environments [1-5]. However, only a few works focus on the CP MS wearable antennas [1-3, 5-7].

Several Circularly polarized (CP) antennas have been studied, including microstrip patch antennas [31], filtering antennas, and anisotropic artificial ground plane loaded antennas [32]. The truncated corner of the microstrip patch antenna in [33] are intended to produce CP unidirectional radiation. However, the coaxial probes that feed these antennas make them difficult to use in wearable applications. A wearable biotelemetric system with a tiny CP antenna integrated with band-pass filter circuits is proposed in [31]. However, it uses a rigid substrate that cannot conform to the human body.  In [31], a flexible CP antenna operating at industrial, scientific, and medical (ISM) band 2.4 GHz is investigated. However, these wearable CP antennas have a limited axial ratio (AR) bandwidth, which could cause performance degradation due to a potential frequency offset.”

  1. The suggested first reference is added as Ref. [1] in the revised manuscript as Al-Gburi, A. J. A.; Zakaria, Z.; Alsariera, H.; Akbar, M. F.; Ibrahim, I. M.; Ahmad, K. S.; Ahmad, S.; Al-Bawri, S. S. Broadband Circular Polarised Printed Antennas for Indoor Wireless Communication Systems: A Comprehensive Review. Micromachines 2022, 13 (7), 1048. https://doi.org/10.3390/mi13071048.
  2. The suggested second reference is added as Ref. [41] in the revised manuscript as Mahmood, S. N.; Ishak, A. J.; Saeidi, T.; Soh, A. C.; Jalal, A.; Imran, M. A.; Abbasi, Q. H. Full Ground Ultra-Wideband Wearable Textile Antenna for Breast Cancer and Wireless Body Area Network Applications. Micromachines2021, 12 (3), 322. https://doi.org/10.3390/mi12030322.

Comments 2:   Since the antenna is applied for Microwave imaging applications, I want to see the Specific Absorption Rate (SAR) calculation weights in terms of 1 gram up to 10 grams of tissue?

Response: Specific Absorption Rate (SAR) at different frequency is shown in following figure X and calculation weights in terms of 1 gram up to 10 grams of tissue is added in Sec 4.3 and compared in Table 2 in the revised manuscript as follows: “The Specific Absorption Rate (SAR) can determine how much power of these radiators acting as sensors are absorbed by the human tissues. The antenna can be called safe if its maximum SAR value didn’t exceed 1.6 W/Kg based on the IEEE C95.3 standard. The simulated Specific Absorption Rate (SAR) is calculated using CST simulator at 2 GHz and 5 GHz. SAR values at 100 mW transmitted power of 2.32 and 0.98 W/Kg, respectively are recorded for 1 g and 10 g of tissue standards respectively. For further lower transmitted power of 50 mW, SAR level of 1.163 W/Kg and 0.489 W/Kg power are recorded for for 1 g and 10 g.  This falls in  the safe zones for SAR of 1.6 W/Kg and 2 W/Kg for 1g and 10 g respectively. Morever, The SAR level for the prposed sensor is measured in SAR Lab at electronics research institute as shown in Fig. 22. Measured SAR levels are recorded in Table 2, at different power levels of 5,10,15 and 20 dBm. SAR values for the proposed sensor maintain safe levels at both 1g and 10 g standarads.”

Table 2.  Comparison of measured SAR levels for the Circularly Polarized Monopole Sensor.

SAR level for CPMS

Power level (dBm)

10g

1g

0.07

0.201

5

0.086

0.293

10

0.211

0.594

15

0.333

0.944

20

(a)

(b)

(c)

(d)

Figure X. SAR calculation by using CST simulator at different frequency (a) 1.8 GHz, (b) 2.1 GHz, (c) 2.45 GHz and (d) 5.2 GHz.

Comments 3: It needs to be evident how the proposed design can be used to address the research gaps of the current study. Significant improvements need to be made by the authors to emphasize the significance of this paper. For example, how many percentages of Axial Ratio improvements are obtained by the CP proposed design?

Response: Percentage of Axial ratio has been added to Comparison table in the manuscript (Table 3). It could be deduced from the table that the proposed circularly polarized antenna-based sensor realized 73% axial ratio percentage (2.2 GHz). This is relatively higher than other circularly polarized antennas presented in literature. This high axial ratio is recorded with good average gain of 6 dBi and broad band of 2.2 GHz.

  1. Conclusion is updated in the manuscript as follows: “Table 3 shows that most of published textile antennas are linear polarized [3, 40-43, 46, 47] with limited bandwidth operation. For SAR calculations, most of published textile-based sensor acquire low SAR at low transmitted power levels. The proposed sensor acquires low SAR of 0.489 W/Kg at 50 mW. Proposed CPMS acquire higher gain of 6 dBi wider bandwidth and axial ratio percentage of 73% compared to another CP antennas reported in literature [2, 5-7, 45]. The proposed textile microwave sensor acquires wide bandwidth, moderate gain and circularly polarized output with small footprint and safe SAR values. All these aspects make the proposed sensor is good candidate for using it in the wireless body area network (WBAN) and breast cancer imaging. Multiple copies of this sensor could be integrated within women bra to monitor breast in the comfort of their homes. From the previous it could be indicated that the proposed CPMS is very suitable for breast cancer detection and wearable garments as smart bra for different breast cancer detection and biomedical applications.”

  1. Updated Table 3 is shown as follows:

Table 3 Performance Comparison between proposed antenna sensor and previously reported architectures.

BW of AR (GHz)

CP/LP

SAR (W/Kg) in 10g

Fabrications

Dimensions (W×L×h) mm3

Gain (dBi)

BW(GHz)

Ref.

2.39–2.57 (7%)

CP

81.5@(1W)

Rogers 3010

9.2 × 9.2 × 1.27

NM

2.39-2.57

2

-

LP(MIMO)

1.93

Denim and felt

24×24×1.4

5.72

4.8-30

3

5.2-7.1

(31%)

CP

367.86 mW @(at 2 W)

Felt

32:5 × 42 × 1

5.7

3.6–13

5

2.3-2.864 (22%)

CP

NM

Felt

76× 76×1.5

4.9

1.086

6

5.730–5.955

(4%)

CP

0.294

Flexible Panasonic R‐F770

35 ×35 ×2.24

7.2

5.67–6.05

7

-

LP

NM

Cordura

80× 30 ×0.5

2.01

0.9,1.8

40

-

LP

0.09

Denim

60 × 50 ×0.7

10.5

7-28

41

-

LP

NM

FR4

27×27 ×1.6

8.65

1.23-20

42

-

LP

3.28×10-6

Felt

36 ×18×3

7.81

3.1-6.5

43

-

Dual Polarized

NM

Kapton Polymidie

20 ×20 ×0.05

NM

2-4

44

2.38–2.5 1(5%)

CP

242

RT Duroid

10 × 10 × 0.3

-7.79

2.3-2.64

45

-

LP

0.0014@ (1mW)

Polyster fabric

70 × 50 × 0.5

2.9

1.198-4

46

-

LP

1 (0.5W)

Flexible Kapton Polyimide

13×13×0.125

4.4

8.6-14

47

1.8-4 (73%)

CP

0.489 ( 50 mW)

Cotton

33.5× 33.5 ×2

6

1.8-8

This wok

NM(not mention in the paper), CP: Circular Polarized, LP: Linear Polarized.

Comments 4: I would suggest adding a brief paragraph to discuss the physical feasibility of the proposed method: why cancer can be detected by microwaves? Why cancer cells are electromagnetically so different from normal cells?

Response: The physical feasibility of the proposed method using Microwave to detect cancer and why cancer cells are electromagnetically so different from normal cells is addressed in the revised manuscript as follows:

“Microwave detection techniques provide a safer approach as it uses non-ionizing radiation.  Microwave imaging relies on detecting differences in electrical properties between tumor and normal tissues within the breast [11]. The contrast ratio of dielectric properties of normal breast tissue to malignant tissue is close to 10 as reported in [12]. Hence, the microwave signal can distinguish between the healthy and the tumorous tissues. Several Microwave-based detection systems have been reported in [9, 10, 13] for cancer detection, most of them focused on planar structures. Development of wearable microwave imaging system is continuously seeking compactness, robustness, mobility, cost efficiency and non-intrusive features. Realizing these features allows regular, long-term assessment for cancer screening conditions as well as the encompassing environment.”

Comments 5:   What impact does the monopole antenna in the proposed wearable antenna have on the circularly polarized technique at 1.8 GHz to 8 GHz?

Response: The proposed textile microwave sensor acquires wide bandwidth, moderate gain and circularly polarized output with small footprint and safe SAR values. All these aspects make the proposed sensor is good candidate for using it in the wireless body area network (WBAN) and breast cancer imaging. Multiple copies of this sensor could be integrated within women bra to monitor breast in the comfort of their homes.  

The given response has been added to the Conclusion as follows “

Comments 6: The quality of the figures presented in this paper could be better, and the fonts need to be more significant to read. Please improve the quality of the figures, especially for Figure 12.

Response: The quality of the figures and the fonts in the proposed manuscript have been improved as well as Figure 12.

Comments 7: The differences between the simulated results and the measured results in Figure 21 should be discussed more.

Response: A paragraph is added in the revised manuscript version that explained the difference of the results between simulated and measured as follow: “The differences between simulated and measured results at low and high frequencies could be attributed to several reasons. First, the effects of the coaxial cable used in the measurement. Second, the conductor fabric and cotton substrate are uneven and there is air between the layers. Third, soldering SMA connector with silver paste to the textile feeder line. Fourth, other electromagnetic interference signals in the atmosphere and ideal model used in simulations as well as manufacturing and measurement tolerances in the positioned antenna.”

Comments 8: References in Table 3 are incorrect; please correct them!

Response: The References in Table 3 have been corrected in the revised manuscript and some references are added.

Comments 9: The Conclusions should be rewritten with the updated results above. The authors are required to revise the comments mentioned above carefully.

Response: We deeply appreciate the reviewer for his/her positive evaluation and valuable comments. According to reviewers’ comments, we have revised the manuscript and improve the figures quality.

  1. A paragraph has been added to the Conclusion section as follows “Table 3 shows that most of published textile antennas are linear polarized [3, 40-43, 46, 47] with limited bandwidth operation. For SAR calculations, most of published textile-based sensor acquire low SAR at low transmitted power levels. The proposed sensor acquires low SAR of 0.489 W/Kg at 50 mW. Proposed CPMS acquire higher gain of 6 dBi wider bandwidth and axial ratio percentage of 73% compared to another CP antennas reported in literature [2, 5-7, 45]. The proposed textile microwave sensor acquires wide bandwidth, moderate gain and circularly polarized output with small footprint and safe SAR values. All these aspects make the proposed sensor is good candidate for using it in the wireless body area network (WBAN) and breast cancer imaging. Multiple copies of this sensor could be integrated within women bra to monitor breast in the comfort of their homes. From the previous it could be indicated that the proposed CPMS is very suitable for breast cancer detection and wearable garments as smart bra for different breast cancer detection and biomedical applications.”

  1. Updated Table 3 is shown as follows with Axial Bandwidth values added to the Table of comparison:

Table 3 Performance Comparison between proposed antenna sensor and previously reported architectures.

BW of AR (GHz)

CP/LP

SAR (W/Kg) in 10g

Fabrications

Dimensions (W×L×h) mm3

Gain (dBi)

BW(GHz)

Ref.

2.39–2.57

(7%)

CP

81.5@(1W)

Rogers 3010

9.2 × 9.2 × 1.27

NM

2.39-2.57

2

-

LP(MIMO)

1.93

Denim and felt

24×24×1.4

5.72

4.8-30

3

5.2-7.1

(31%)

CP

367.86 mW @(at 2 W)

Felt

32:5 × 42 × 1

5.7

3.6–13

5

2.3-2.864 (22%)

CP

NM

Felt

76× 76×1.5

4.9

1.086

6

5.730–5.955

(4%)

CP

0.294

Flexible Panasonic R‐F770

35 ×35 ×2.24

7.2

5.67–6.05

7

-

LP

NM

Cordura

80× 30 ×0.5

2.01

0.9,1.8

40

-

LP

0.09

Denim

60 × 50 ×0.7

10.5

7-28

41

-

LP

NM

FR4

27×27 ×1.6

8.65

1.23-20

42

-

LP

3.28×10-6

Felt

36 ×18×3

7.81

3.1-6.5

43

-

Dual Polarized

NM

Kapton Polymidie

20 ×20 ×0.05

NM

2-4

44

2.38–2.5

1(5%)

CP

242

RT Duroid

10 × 10 × 0.3

-7.79

2.3-2.64

45

-

LP

0.0014@ (1mW)

Polyster fabric

70 × 50 × 0.5

2.9

1.198-4

46

-

LP

1 (0.5W)

Flexible Kapton Polyimide

13×13×0.125

4.4

8.6-14

47

1.8-4

(73%)

CP

0.489 ( 50 mW)

Cotton

33.5× 33.5 ×2

6

1.8-8

This wok

NM(not mention in the paper), CP: Circular Polarized, LP: Linear Polarized.

Reviewer 3 Report

This paper on antennas for breast cancer detection raises many questions that I note below:

Why is it so important to ahve an AR bandwidth on this as surely circular polarisation does not matter so much for this application? Rather we would want dual polarisation as the AR will not tell us much about the antenna coupling EM into the breast.

Why are we interested in the antenna pattern and AR when on the breast as these are sensors and not antennas for communication from the breast, which would be the job of a radio node it would be connected to? Thus really the work here is to be how the sensors can detect tumour and would be with a specific bra worn for detection for a short time for use domestically, not to be worn all the time. This would form a non invasive regular use detector connected to a handheld device for use that could make early stage alerts to then seek further more detailed medical attention.

Why are we comoparing silver fabric with copper conductor? At the end of the day they are both conuctors and make little difference to the antenna.

The results from Figs. 13 to 16 are most crucial since it is there where we need to demonstrate the capability to detect tumour that has only been carried out in simulation and not measurement while also for a single scenario of tumor in the middle with different size. But what about position? There really does need to be more exhaustive set of results to justify the work in this paper. Also is the reflection coefficient of the two antennas the same in Fig. 15 so only one is plotted? We need a clear metric that shows this can really work and it is concerning that the simulation and real scenario would have too much uncertainty that it is very hard to distinguish if in fact these sensors ever really could detect tumours in a real bra on real breasts. Substantial repeatability would be required there that I cannot honestly have strong confidence in the results given here while also there is a lot of irrelevant information in the paper. More rigorous analysis of breast voxel models is required with the insertion of tumour to demonstrate the capability.

I note that non cited works such as the following are more advanced in this regard:

Mahmood, S.N.; Ishak, A.J.; Jalal, A.; Saeidi, T.; Shafie, S.; Soh, A.C.; Imran, M.A.; Abbasi, Q.H. A Bra Monitoring System Using a Miniaturized Wearable Ultra-Wideband MIMO Antenna for Breast Cancer Imaging. Electronics 2021, 10, 2563. https://doi.org/10.3390/electronics10212563

E. Porter, H. Bahrami, A. Santorelli, B. Gosselin, L. A. Rusch and M. Popović, "A Wearable Microwave Antenna Array for Time-Domain Breast Tumor Screening," in IEEE Transactions on Medical Imaging, vol. 35, no. 6, pp. 1501-1509, June 2016, doi: 10.1109/TMI.2016.2518489.

Thus I am sorry I cannot support this paper due to clear shortcomings in its technical soundness while novelty appears limited. I would note some other minor feedback as follows:

It's not good to have sub headings under section headings like in section 2. They should have some text of one or two sentences first to keep the flow of the paper.

Jeans are spelt this way, not Jenes. Also Captin is not a fabric and I'm not sure what was meant here. Possbbly kapton but that is not a textile so would not make sense.

Author Response

Response Letter

Manuscript ID: micromachines-2181361

Manuscript Title: Circularly Polarized Textile Sensors for Microwave-based Smart Bra
Monitoring System

Authors: * Dalia N. Elsheakh *, Yasmine K. Elgendy, Mennatullah E. Elsayed,
Angie R. Eldamak

First, we would like to sincerely thank the editor and the reviewers for their valuable comments. We have considered all the reviewers’ comments and suggestions and have modified our manuscript accordingly.

We have also prepared detailed responses for reviewers’ comments. In the revised paper, we use highlighted yellow text for modified sections, figures & tables in the paper based on the reviewers’ comments.

Sincerely yours,

Comments of Reviewer 3

Comments 1: Why is it so important to have an AR bandwidth on this as surely circular polarisation does not matter so much for this application? Rather we would want dual polarisation as the AR will not tell us much about the antenna coupling EM into the breast.

Response: We added a paragraph to emphasis designing CP antenna and recording the AR  in the revised version in the Introduction as follows: “Wearable antennas can bend and stretch while in use and accordingly the radiation properties are varied. Thus, CP antenna is favoured in developing flexible wearable sensors and systems. CPMS microwave sensor waves contain both vertical and horizontal electric field components. Thus, receiver antennas can pick up radiation waves from a variety of angles; when these waves are reflected or refracted. Moreover, circular polarization (CP) antennas reduce polarization mismatch, flexibility in terms of orientation and increased mobility. Moreover, it is a good choice for encountering multipath fading and establishing reliable channels when compared to linearly polarized (LP) antennas, especially in crowded interior environments [1-5]. However, only a few works focus on the CP MS wearable antennas [1-3, 5-7].”

“The proposed circularly polarized sensor is continuation to the work in [30] to develop Smart Bra as shown in Fig. 1. In [30], the textile-based sensor is presented as linearly polarized microstrip antenna and has an impedance bandwidth from 1.8 to 2.4 GHz and from 4 up to 10 GHz at |S11| ≤ −10 dB. By enabling circularly polarized output, a new monitoring indicator, axial ratio, is added. Axial ratio will be recorded in addition to S-parameters (|S11| for reflection and |S21| for transmission). The proposed CPMS has a broad axial ratio (AR) bandwidth that allows efficient operation in challenging conditions and in proximity to lossy, dispersive, and heterogeneous human tissues. Moreover, it will be also used in analyzing data measured from breast. This will increase the samples of collected data and will contribute to increasing the detection accuracy. The proposed system is meant to help women in early detection and continuous monitoring for breast in the comfort of their homes.”

Comments 2:  Why are we interested in the antenna pattern and AR when on the breast as these are sensors and not antennas for communication from the breast, which would be the job of a radio node it would be connected to? Thus really the work here is to be how the sensors can detect tumour and would be with a specific bra worn for detection for a short time for use domestically, not to be worn all the time. This would form a non-invasive regular use detector connected to a handheld device for use that could make early-stage alerts to then seek further more detailed medical attention.

Response: Radiation pattern and gain are very important parameters to describe the antenna for WBAN as well. This has been added to the revised manuscript as follows:

  1. In the Introduction: “In this paper, we introduce an ultra-wideband (UWB) antenna that can be utilized as sensors in two major applications: (1) wireless body area networks (WBANs) and (2) breast cancer imaging. This paper presents a fully textile CPW monopole antenna with operating bandwidth from 1.8 GHz to 8 GHz and circularly polarized output in the band of 1.8 - 4 GHz.”
  2. In the Introduction, “A wireless body area network (WBAN) is a group of low-power, miniature, light-weight wireless sensors that monitor internal and external body health processes. WBANs help in monitoring physiological behavior patterns in humans and studying disease prevention and control [1]. The WBAN's ultra-lightweight, wearable sensors can be either off-body or on-body [2] and could be operated in MIMO systems [3-5]. Numerous applications of WBAN have embraced several devices on a large-scale including brain recording, glucose monitoring, intracranial pressure monitoring as well as breast cancer detection [2, 6]. WBAN use wearable, light weight antennas as a feature to transmit and receive signals to the human body. In this paper a wearable bra is proposed to monitor and scan breast cancer. This Smart Bra is presented as a component of a WBAN system such that women won't need to visit the hospital regularly [7].
  3. In the Introduction, The advantage of circularly polarization has been highlighted in Introduction as follows: “Wearable antennas can bend and stretch while in use and accordingly the radiation properties are varied. Thus, CP antenna is favored in developing flexible wearable sensors and systems. CPMS microwave sensor waves contain both vertical and horizontal electric field components. Thus, receiver antennas can pick up radiation waves from a variety of angles; when these waves are reflected or refracted. Moreover, circular polarization (CP) antennas reduce polarization mismatch, flexibility in terms of orientation and increased mobility. Moreover, it is a good choice for encountering multipath fading and establishing reliable channels when compared to linearly polarized (LP) antennas, especially in crowded interior environments [1-5]. However, only a few works focus on the CP MS wearable antennas [1-3, 5-7].

Several Circularly polarized (CP) antennas have been studied, including microstrip patch antennas [31], filtering antennas, and anisotropic artificial ground plane loaded antennas [32]. The truncated corner of the microstrip patch antenna in [33] are intended to produce CP unidirectional radiation. However, the coaxial probes that feed these antennas make them difficult to use in wearable applications. A wearable biotelemetry system with a tiny CP antenna integrated with band-pass filter circuits is proposed in [31]. However, it uses a rigid substrate that cannot conform to the human body.  In [31], a flexible CP antenna operating at industrial, scientific, and medical (ISM) band 2.4 GHz is investigated. However, these wearable CP antennas have a limited axial ratio (AR) bandwidth, which could cause performance degradation due to a potential frequency offset.”

  1. The suggested first reference is added as Ref. [1] in the revised manuscript as Al-Gburi, A. J. A.; Zakaria, Z.; Alsariera, H.; Akbar, M. F.; Ibrahim, I. M.; Ahmad, K. S.; Ahmad, S.; Al-Bawri, S. S. Broadband Circular Polarised Printed Antennas for Indoor Wireless Communication Systems: A Comprehensive Review. Micromachines 2022, 13 (7), 1048. https://doi.org/10.3390/mi13071048.
  2. The suggested second reference is added as Ref. [41] in the revised manuscript as Mahmood, S. N.; Ishak, A. J.; Saeidi, T.; Soh, A. C.; Jalal, A.; Imran, M. A.; Abbasi, Q. H. Full Ground Ultra-Wideband Wearable Textile Antenna for Breast Cancer and Wireless Body Area Network Applications. Micromachines2021, 12 (3), 322. https://doi.org/10.3390/mi12030322.

Comments 3: Why are we comparing silver fabric with copper conductor? At the end of the day they are both conductors and make little difference to the antenna.

Response: Simulations are conducted using two types of conductors as 0.5 mm copper clad conductor and sliver conducting fabric (sheet resistance 0.5 Ω/? ). These simulations validate similar performance of the proposed monopole antenna using conductor fabric compared to traditional copper clad. This is conducted to validate the coductive textiles can replace tranditinal conductor clad while having same performance. Both types of conductors reveal similar operation in terms of reflection coefficient, impedance matching (real and imaginary) and gain shown in Fig. 7(a)-(c) respectively. The gain using both conductors shown in Fig.7(c). Gain values are almost the same in the range 3.5 to 6 GHz with maximum realized gain of 8 dBi by using copper conductor while the gain is reduced to 5 dBi by using textile conductor fabric.

Comments 4: The results from Figs. 13 to 16 are most crucial since it is there where we need to demonstrate the capability to detect tumour that has only been carried out in simulation and not measurement while also for a single scenario of tumor in the middle with different size. But what about position?

Response: The effect of varying position has been added to the revised manuscript and Fig.15 as follows: “Moreover, the location of tumor is varied and the simulated results of the tumor shifted from the center (Z: offset distance from center presented in Fig.10) is also studied as shown in Fig. 15. Fig. 15 shows the rate of change of the properties of the CPMS is high when the tumor is on the same line between the sensors, and when the tumor is farther from the line between the two sensors, the effect of these changes decreases. The magnitude of the reflection and transmission coefficient are changed by more than 30 dB when the antenna”

There really does need to be more exhaustive set of results to justify the work in this paper. Also is the reflection coefficient of the two antennas the same in Fig. 15 so only one is plotted?  We need a clear metric that shows this can really work and it is concerning that the simulation and real scenario would have too much uncertainty that it is very hard to distinguish if in fact these sensors ever really could detect tumours in a real bra on real breasts. Substantial repeatability would be required there that I cannot honestly have strong confidence in the results given here while also there is a lot of irrelevant information in the paper. More rigorous analysis of breast voxel models is required with the insertion of tumour to demonstrate the capability.

Response: Repeated test of the proposed CPMS has been conducted. This has been validated by measuring the reflection coefficient values versus frequency for three copies of the same design. The measurements are shown in the following figure. Most of these copies of the CPMS sensor operated in the required band. The difference of these results could be attributed from the soldering of sliver paste. Additionally, compared to traditional printed circuit board (PCB) processes, textile-based systems' manufacturing tolerances are much looser, resulting in a significantly greater range of uncertainty.

(a)

(b)

(c)

Figure 1.  Simulated of proposed CPMS parameters with tumor location (second scenario shown in Fig.10 (b)) (a) |S11| (b) |S21| in dB and (c) Axial ratio.

Figure 2.  The reflection coefficient values versus frequency for three copies of the proposed microwave sensors.

Comments 5:   I note that non cited works such as the following are more advanced in this regard:

Mahmood, S.N.; Ishak, A.J.; Jalal, A.; Saeidi, T.; Shafie, S.; Soh, A.C.; Imran, M.A.; Abbasi, Q.H. A Bra Monitoring System Using a Miniaturized Wearable Ultra-Wideband MIMO Antenna for Breast Cancer Imaging. Electronics 2021, 10, 2563. https://doi.org/10.3390/electronics10212563
E. Porter, H. Bahrami, A. Santorelli, B. Gosselin, L. A. Rusch and M. Popović, "A Wearable Microwave Antenna Array for Time-Domain Breast Tumor Screening," in IEEE Transactions on Medical Imaging, vol. 35, no. 6, pp. 1501-1509, June 2016, doi: 10.1109/TMI.2016.2518489.

Response: We added these references with other references as [2] and [3], respectively. 

Comments 6: Thus I am sorry I cannot support this paper due to clear shortcomings in its technical soundness while novelty appears limited. I would note some other minor feedback as follows:It's not good to have sub headings under section headings like in section 2. They should have some text of one or two sentences first to keep the flow of the paper.

Response: Some text of one or two sentences have been added before subsections in Section 2, Section 3 and Section 4 to keep the flow of the paper.

  1. Section 2: “In this section, the design of the proposed circularly polarized microwave sensor (CPMS) is presented. This includes presenting materials and design steps for the proposed sensors. In Section 2.1, characterization of different conductive and substrates materials will be illustrated. This involve presenting electrical properties of different materials used in sensor fabrication. In Section 2.2, design steps for the proposed sensor will be shown as well as its main parameters. In Section 2.3, different models and phantoms for breast tissues will be discussed. Electrical properties of fabricated breast phantoms will be measured and presented in Section 2.3. The fabricated phantoms will be used to assess the operation of the proposed sensors in proximity to breast tissues.
  2. Section 3: “In this section, simulation results for proposed sensor in air and with breast and tumor phantoms will be presented. All simulation results are recorded using CST studio suite as well as high frequency structure simulator (HFSS). In Sec 3.1, the effect of changing conductor material will be studied. Conductor is an essential material to fabricate radiator and ground plane for the proposed antennas. Simulated reflection coefficient, impedance and gain using two conductor types will be compared in Sec 3.1. In Sec 3.2, simulation of proposed sensors with breast phantoms with and without tumor will be presented.”
  3. Section 4: “This section will present fabrication and measurements of the proposed CPMS. Experimental results will be compared to simulations. In Sec 4.1, Measurements in air and off-body will be presented. This includes measurements of magnitude and phase for reflection coefficients, transmission coefficients, gain as well as Axial ratio versus frequency. Moreover, Measurements for proposed sensor on breast phantoms are performed and analyzed. SAR levels are calculated and presented in Sec. 4.2.”
  4. Sub Section 4.1 and 4.2 are merged into one Section.

Comments 7: Jeans are spelt this way, not Jenes. Also Captin is not a fabric and I'm not sure what was meant here. Possibly kapton but that is not a textile so would not make sense.

Response:  The word “Jeans” has been corrected and the typo of “Captain” is changed to Crepe as referring to Crepe Fabric.

Round 2

Reviewer 1 Report

As the authors have well addressed all my comments in the revision of micromachines-2181361, the present version can be recommended to be published in Micromachines.

Author Response

Thank the editor and the reviewers for their valuable comments. We have considered all the reviewers’ comments and suggestions and have modified our manuscript accordingly.

Reviewer 2 Report

The authors have revised the given comments successfully, and I believe the article is ready now to be published in a reputational journal like Micromachines. However, there are still typos and spacing errors that need to be carefully checked. Best regards

Author Response

Thank you very much for your effort in reviewed our manuscript; according to your 

Reviewer 3 Report

I am still unfortunately very unhappy with this paper and regret I cannot support it. The authors have come back too quickly on this matter. For this to be useful to actually show a bra could be used for tumour detection it is clear from the results that the maximum frequency to go to is 4GHz and so that band should be focused on. Then careful analysis with both the reflection coefficient and to determine the best location for two sensors on a bra to measure transmission coefficient is then necessary. This is clearly how the work should be carried out to solve the research problem. Authors have gone using AR and gain that means nothing as that is only relevant to the antenna being used for WBAN mode but that isn't going to be relevant. This would be designed in a system where the user would wear the bra for a short time connected in proximity to a reader device by a cable. There is no need to use this in a normal bra worn all day. It is purely a domestic device. This therefore rules out all far field aspects of the sensors and then must on showing rigorously how it will detect tumour in a breast. This has not been carried out and the best way to enable this detection is to form a correlation metric comparing the s11, s12 and s11 up to 4GHz with a non tumour case to then evaluate detection reliability. Also this must matter to any size breast as well as position of tumour and this has not been worked out. therefore I am sorry I cannot support this work.

Author Response

Response Letter

Manuscript ID: micromachines-2181361

Manuscript Title: Circularly Polarized Textile Sensors for Microwave-based Smart Bra
Monitoring System

Authors: * Dalia N. Elsheakh *, Yasmine K. Elgendy, Mennatullah E. Elsayed,
Angie R. Eldamak

First, we would like to sincerely thank the editor and the reviewers for their valuable comments. We have considered all the reviewers’ comments and suggestions and have modified our manuscript accordingly.

We have also prepared detailed responses for reviewers’ comments. In the revised paper, we use highlighted yellow text for modified sections, figures & tables in the paper based on the reviewers’ comments.

Sincerely yours,

Comments of Reviewer 3

Comments 1:I am still unfortunately very unhappy with this paper and regret I cannot support it. The authors have come back too quickly on this matter. For this to be useful to actually show a bra could be used for tumour detection it is clear from the results that the maximum frequency to go to is 4GHz and so that band should be focused on.

Response: We would like to thank the reviewer for his/her feedback. We designed a wide-band antenna for designated application in the paper to study detection and sensing problem over a wide band. From the first round of tests and validation, we agree with the reviewer that lower side of band is better and show more sensitivity in detection. This shows clearly also in studying the effect of varying position of tumour updated in the submitted manuscript. This manuscript is part of project, we are working on and further work is conducted and will be show up in future publications.

Comments 2:Then careful analysis with both the reflection coefficient and to determine the best location for two sensors on a bra to measure transmission coefficient is then necessary. This is clearly how the work should be carried out to solve the research problem.

Response: This manuscript is part of project; we are working on, and further work is conducted and will be show up in future publications. Implementing two antenna elements is a step towards recording transmission via Breast. This is intended to validate if transmission properties will detect changes by presence of Cancerous cells using textile-based sensors. However, from literature background and our work, it is not enough to use two sensors on each side for a bra to detect tumour in Breast. More sensors should be placed and oriented on the Bra to cover and scan whole breast. Currently we are performing four antenna elements and testing feasibility of adding more elements to our system.

Comments 3: Authors have gone using AR and gain that means nothing as that is only relevant to the antenna being used for WBAN mode but that isn't going to be relevant. This would be designed in a system where the user would wear the bra for a short time connected in proximity to a reader device by a cable. There is no need to use this in a normal bra worn all day. It is purely a domestic device. This therefore rules out all far field aspects of the sensors and then must on showing rigorously how it will detect tumour in a breast. This has not been carried out and the best way to enable this detection is to form a correlation metric comparing the S11, S12 and S11 up to 4GHz with a non tumour case to then evaluate detection reliability.

Response: We would like to thank the reviewer for his/her Feedback. Far field properties such as pattern  show up in the revised manuscript to support adding WBAN application requested by one of the reviewers (reviewer 2 in previous rounds).

For Breast Cancer detection, it is operating in proximity to human body where correlation metric comparing S11, S12 (magnitude and phase) with and without tumor is conducted and analysed for detection problem.

Having circularly polarized and using AR as additional detection parameter has been also referred in other publications for Wearable applications:

  1. Kumar, S.; Nandan, D.; Srivastava, K.; Kumar, S.; Singh, H.; Marey, M.; Mostafa, H.; Kanaujia, B. K. Wideband Circularly Polarized Textile MIMO Antenna for Wearable Applications. IEEE Access20219, 108601–108613. https://doi.org/10.1109/access.2021.3101441.
  2. Lui, K. W.; Murphy, O. H.; Toumazou, C. A Wearable Wideband Circularly Polarized Textile Antenna for Effective Power Transmission on a Wirelessly Powered Sensor Platform. IEEE Transactions on Antennas and Propagation 2013, 61 (7), 3873–3876. https://doi.org/10.1109/tap.2013.2255094.
  3. Saha, P.; Mitra, D.; Parui, S. K. A Circularly Polarized Implanted Monopole for Biomedical Applications. Progress In Electromagnetics Research C201885, 167–175. https://doi.org/10.2528/pierc18051807.

As for Wearable application, realized gain and efficiency are measured and added to the revised manuscript as Fig. 20. The text has been updated as follows:

“Finally, the proposed CPMS gain, and radiation efficiency are measured in the free space in an Anechoic chamber in Microwave lab at Faculty of Engineering, Ain shams University. The measurement system is composed of Near-field Systems, Inc. 700S-30, with one wideband double-ridged horn antenna and NSI-RF-RGP-10, connected to a Vector Network Analyzer, Rohde & Schwarz ZVB14) as shown in Fig. 20(a). Fig. 20(b) shows that the realized gain of the antenna in free space is around 4 dBi over the operating band. The efficiency is also calculated with acceptable level of 60%. Efficiency is an important parameter to be recorded for microwave-based sensors in WBAN or sensing systems attached to human phantoms.”

Comments 4: Also this must matter to any size breast as well as position of tumour and this has not been worked out. Therefore, I am sorry I cannot support this work.

Response: We would like to thank the reviewer for his/her Feedback. The effect of the size of the breast will be the scope of future of publications. Position of the tumor has been studied and added in the revised manuscript as shown in Fig.10, Fig.11, Fig.12, Fig.13, Fig.14 and Fig. 15.

Round 3

Reviewer 3 Report

I have not supported this paper and still do not.